# Trajectory simulation of multi-body parachute system for airdrop-capable UAVs based on fluid-structure interaction

Hanxu Guo[1,2,3], Ziang Gao[1,2,3‡], Zijian Zhu[1,2,3‡], Miao Zhang[1,2,3‡], Jian Zhang [1,2,3‡]*

1 Institute of Engineering Thermophysics, Chinese Academy of Sciences, Beijing, China, 2 School of Aeronautics and Astronautics, University of Chinese Academy of Sciences, Beijing, China, 3 National Key Laboratory of Science and Technology on Advanced Light-duty Gas-turbine, Beijing, China

‡ ZG, ZZ, MZ, and JZ also contributed equally to this work.
* zhangjian@iet.cn

## Abstract

Due to their demonstrated advantages of precision, efficiency, and low cost in disaster relief and commercial logistics, airdrop-capable unmanned aerial vehicle (UAV) are rapidly becoming pivotal tools in modern delivery systems. This paper proposes a novel airdrop-capable UAV with foldable wings. To address the requirements for high-precision deployment and parachute cut-off, a 10-degree-of-freedom (10-DOF) multibody dynamics model of the parachute-UAV system is established based on Kane's equations. The solution process incorporates sixth-order vibration equations to characterize the system's rigid-flexible coupling effects, precisely capturing the motion trajectories under varying initial deployment parameters (initial velocity, parachute diameter). To comparatively analyze the trajectory curves derived from fluid-structure interaction (FSI) simulation and to validate the model's effectiveness, this paper constructs a co-simulation framework. This framework couples Gamma-Theta transition model-based FSI with LS-DYNA to simulate the airdrop dynamics across multiple operating conditions. This study acquires the parachute jettison coordinates of an airdrop UAV under varying deployment parameters, elucidating their parametric dynamic coupling on airdrop trajectories and separation point selection methodology. These findings establish both theoretical principles and technical frameworks for precision guidance and flight trajectory control.

## Introduction

Unmanned aerial vehicles (UAVs) possess distinct advantages such as rapid response, flexible deployment, low cost, and reduced personnel risk. In modern emergency rescue, remote area material supply, search and security operations, airdrop-capable UAVs have emerged as a disruptive technological approach [1]. However, traditional dynamic models of parachute-payload systems for airdrop

**Data availability statement:** All relevant data are within the paper and its Supporting information files.

**Funding:** The author(s) received no specific funding for this work.

**Competing interests:** The authors have declared that no competing interests exist.

applications are overly simplified and struggle to accurately describe motion characteristics under complex airflow conditions. This directly leads to low reliability in selecting the "parachute release point" based on coarse models, which amplifies trajectory deviations during the parachute descent phase. This paper innovatively proposes an airdrop-capable UAV that integrates folding wings with parachute airdrop technology, achieving a holistic integration of UAV storage, transportation, and deployment processes. This significantly enhances deployment accuracy, operational efficiency, and coverage range. Nevertheless, the deployment process involves challenges such as trajectory prediction for flexible parachutes and parachute release point selection, which entail complex aerodynamic-structural coupling. Nonlinear factors including turbulence interference, flow separation, spillover effects caused by parachute flexibility, and dynamic attitude oscillations make real-time detection of the system's dynamic equilibrium difficult [2].

Research into parachute systems encompasses a wide array of simulation techniques, ranging from multi-body dynamics to fluid-structure interaction, all aimed at accurately predicting trajectory, dynamic characteristics, and structural behavior.

Ortega et al. [2] addressed deployment phase dynamics with mass-spring-damper techniques and used FEM for structural load calculation. Doherr et al. [3] developed a 9-DOF rotating parachute program to simulate system trajectory and dynamics. Dutta et al. [4] established a multi-body flight dynamics model for supersonic parachutes, achieving trajectory reconstruction and model validation. Su et al. [5] built a multi-body dynamic model using Kane's method, describing motion with generalized speeds and deriving generalized forces and differential equations; this model better reflects nacelle disturbance torque than single-body models. Zhang et al. [6] studied small folding-wing UAVs launched from high-altitude balloons, modeling them as multi-rigid-body connected structures using Kane's method, simulating parachute deployment, and calculating recovery trajectories. Jang et al. [7] investigated the altitude-dependent dynamics of cruciform parachutes. Cao et al. [8] established a 6-DOF rigid-body flight dynamics model for parachute-payload systems, calculating trajectory, attitude, velocity, and impact point parameters, and predicting dispersion using Monte Carlo methods. Gao et al. [9] constructed an FSI model based on multi-body dynamics and LS-DYNA, synchronously simulating parachute opening and trajectory, and analyzing unsteady flow, wake re-contact, and gust effects. Zhu et al. [10] developed a parachute-projectile system model via ALE-based FSI simulation, achieving impact area prediction validated by experiments. Yang et al. [11] designed a trajectory tracking controller using MATLAB simulation, achieving highly consistent simulated trajectories. Zhu et al. [12] proposed a two-way coupled FSI numerical method for parafoil design optimization, improving the prediction accuracy of flexible models. Ghoreyshi et al. [13] studied the flight dynamics of C-17 deployed parachute-payload systems, using CREATE-AV Kestrel for modeling and precise wake field capture. Omar et al. [14] developed a 6-DOF powered parachute model, optimizing trajectory planning with an incremental genetic algorithm, significantly reducing computational cost. Iyer et al. [15] proposed a high-precision, low-computational-cost multi-body parachute-elastic-tether-payload system modeling

method based on Kane's method. Li et al. [16] found parachute flight characteristics determined by the balance of inertia and aerodynamic drag, with inter-parachute interaction most likely at opening. Möller et al. [17] considered the eigenvalue problem of sixth-order ODEs in curved arch vibration models. Zhang et al. [18] established Kane multi-body dynamic models for balloons and parachutes, performing recovery simulation via steady-state spin reduction analysis, and calculating glide trajectories with the Radau pseudospectral method. Gao et al. [19] performed FSI prediction for disk-gap-band parachute inflation using ALE penalty function coupling. Stein et al. [20] proposed a 3D parachute FSI parallel computation strategy, handling large deformations with compatible surface meshes and adaptive meshing. Guglieri et al. [21] effectively presented aircraft terminal deceleration characteristics, revealing simulation output sensitivity to mathematical model complexity. Yu et al. [22] developed a detailed star-shaped folding parachute model, capturing initial inflation load jumps via operator splitting. Sun et al. [23] studied external wind field effects on projectile-parachute terminal trajectories, finding significant impact on landing deviation, with headwinds exacerbating trajectory offset. Shumway et al. [24,25] found spanwise allowance impacts fabric shape, 6° AoA is a vibration turning point, leading edge state dominates FSI, and spanwise prestrain suppresses vibration. Thanh et al. [26,27] found small parachutes have higher oscillation frequencies and are more wind-affected, while larger parachutes might resonate with turbulence during low-altitude descent. The modern physics-embedded approach proposed by Song et al. [28] effectively combines physical laws with data-driven algorithms, demonstrating significant potential in improving computational efficiency and quantifying uncertainties for complex systems.

Existing studies either simplify the parachute as a rigid-body model, neglecting the time-varying fluid-structure interaction effects during actual canopy deployment and descent, which limits trajectory prediction accuracy, or conduct full-stage fluid-structure interaction (FSI) analysis of the parachute airdrop process, resulting in prohibitively high computational costs and excessive resource consumption. Additionally, the selection of parachute release coordinates in related research largely relies on empirical methods, lacking systematic optimization. Moreover, there is insufficient in-depth analysis of the comprehensive impact of key parameters—such as initial ejection velocity and parachute diameter—on the complete airdrop trajectory, making it difficult to support high-precision and highly reliable airdrop operations. In particular, the aircraft must reach a well-defined parachute release point in an attitude-stable state after deployment; otherwise, pull-up risks may arise. Traditional parabolic or steady-state aerodynamic models cannot satisfy the high-fidelity description required for such complex dynamic processes.

To address the above issues, this study abandons the traditional rigid-body assumption and constructs an airdrop dynamics model that incorporates time-varying fluid-structure interaction effects, significantly improving trajectory prediction accuracy under complex aerodynamic conditions. On this basis, the study systematically reveals the coupling mechanisms of multiple parameters, such as initial ejection velocity and parachute diameter, overcoming the limitations of single-variable analysis. In response to the gap in parachute release point decision-making, a trajectory–release point co-optimization method that considers attitude stability constraints is further proposed, achieving a transition from empirical to precisely controllable airdrop operations. This provides theoretical foundations and engineering references for the safety and accuracy of airdrop missions. We have established a "phased multi-fidelity" hybrid simulation framework: during high-risk phases, such as the transient canopy inflation and initial unstable airdrop stage, high-resolution FSI simulations are employed to capture canopy deployment and initial disturbances. During the stable descent phase, the framework switches to a medium-fidelity multibody dynamics model based on Kane's equations. At the final airdrop stage, Monte Carlo analysis is embedded for real-time prediction of the parachute release point.

This study constructs a flexible-rigid coupled dynamics model for the airdrop system, based on multibody system dynamics. By Kane's equations, a ten-degree-of-freedom model encompass the coupling of the parachute, connection mechanism, and UAV is established [9,15,19].Overcoming the limitations of traditional rigid-body assumptions, the small flexible deformations of the parachute are incorporated into the dynamics model via a second-order vibration equation. Aerodynamic load functions are established, thereby dynamically coupling the deformations with the rigid-body motion.

To validate the model's effectiveness, a co-simulation approach employing LS-DYNA explicit dynamics and Computational Fluid Dynamics (CFD) is adopted: The LS-DYNA simulation is utilized for the stage of large flexible deformation during parachute deployment and inflation [9,19]. Subsequently, CFD simulation is employed for the stable deceleration phase characterized by small flexible deformations post-inflation. The Gamma-Theta transition model is integrated within the CFD simulation to obtain the complete airdrop trajectory. Then the results are compared and validated against the multibody dynamics numerical simulations. By comparatively analyzing the effects of different initial deployment velocities and parachute parachute diameters on the system's aerodynamic characteristics, a jettison point selection strategy is ultimately optimized based on specific jettison requirements. This provides theoretical support for the precise guidance and control of the airdrop system.

## Research object

The airdrop-capable UAV employs an aerodynamic configuration featuring a rear-mounted electric propeller, foldable tandem-wing arrangement, and twin vertical tails. The airframe utilizes a carbon fiber composite structure. Control is exclusively implemented via two control surfaces positioned at the left and right trailing edges of the rear wing. UAV pitch control is achieved through symmetric deflection of both control surfaces, while roll control is accomplished via differential deflection. The control surface deflection angle ranges from ±30°, with a response time of ≤50 ms. The derivatives of the pitch, roll, and yaw moment coefficients with respect to the control surface deflection angle are −1.5, 0.28, and 0.06, respectively. Key parameters of the UAV are presented in Table 1.

The aerodynamic configuration of the airdrop-capable folding-wing UAV is depicted in Fig 1. When the UAV decelerates to a safe deployment speed threshold via the parachute system, the torsion spring mechanism triggers release, applying equal-magnitude complementary torques to both wings for smooth and controlled deployment.

**Table 1. Parameters of the airdrop-capable Foldable Wing UAV.**

| Parameter | Value | Value | | Parameter | Parameter | Value |
|---|---|---|---|---|---|---|
| Mass | 8 kg | Length | 0.830m | | Height | 0.115m |
| Canard Span | 1.330m | Main Wing Span | 0.990m | | Vertical Fin Span | 0.188m |
| Canard Area | 0.1064m² | Main Wing Area | 0.0693m² | | Vertical Fin Area | 0.0110m² |

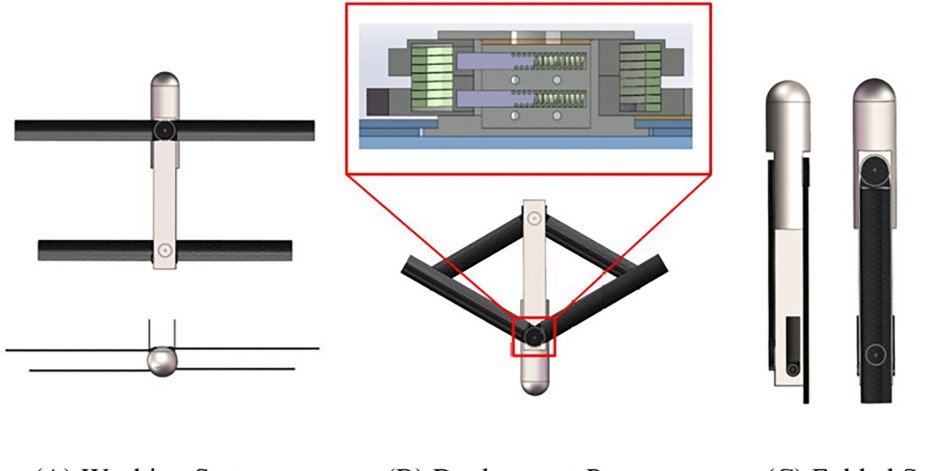

(A) Working State    (B) Deployment Process    (C) Folded State

**Fig 1. Airdrop-capable UAV modeling.**

## Airdrop procedures

The deployment process of the airdrop-capable parachute-UAV system comprises seven distinct phases, as illustrated in Fig 2.

a:   The UAV is stowed inside the launch apparatus. An unmanned or manned carrier aircraft transports the system to the designated maritime area.

b:   The launch apparatus ejects the UAV together with its protective sleeve at approximately 10 m/s using compressed air. After exiting the tube, the parachute deploys. Under the parachute's influence, the UAV body flips end-over-end and subsequently undergoes parabolic motion. During this phase, the sleeve detaches under gravitational force.

c:   During the descent phase, the parachute-UAV system continuously decelerates and stabilizes its attitude angle.

d:   Upon achieving stabilization, the parachute release mechanism activates to sever rigging lines, initiating UAV-parachute separation. Concurrently, wing and empennage restraint straps are jettisoned, enabling full deployment of aerodynamic surfaces.

e:   The UAV's front and rear folding wings achieve full deployment. The tail-mounted propeller commences rotation.

f:   Following predefined control laws, the UAV transitions from a dive state to level cruise flight.

g:   The UAV executes its cruise mission according to the predetermined flight trajectory while maintaining level flight.

## Multi-body dynamics (Kane's method)

The system comprises three components: UAV: Capable of six-degree-of-freedom spatial motion; Link Bearing: Connected to the UAV via a single-DOF revolute joint, permitting only axial relative rotation to regulate roll orientation and maintain upward attitude during airdrop; Parachute: Attached to the link bearing through a three-DOF spherical joint.

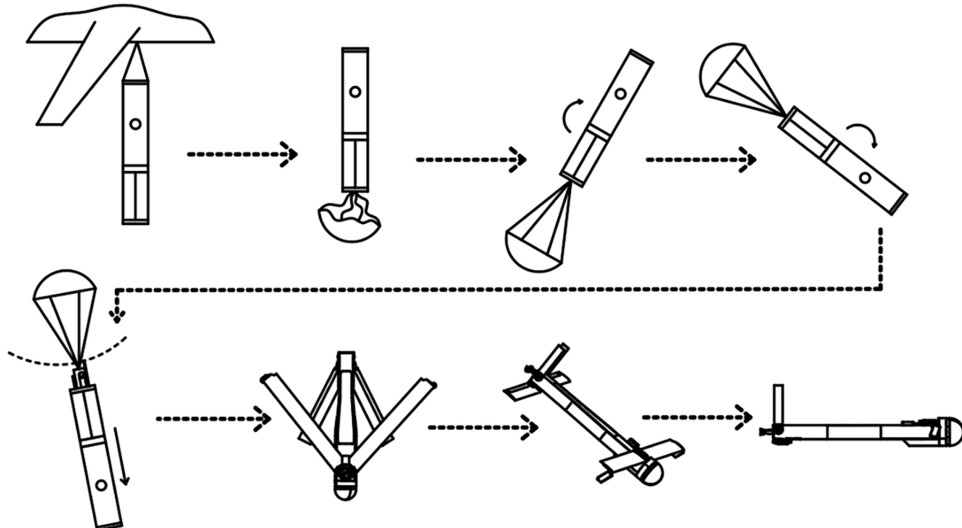

**Fig 2.  Airdrop procedures schematic diagram.**

Kane's methodology computes the position, velocity, and angular velocity vectors of all components. Through the introduction of partial velocities and partial angular velocities, these quantities are unified in the system's centroidal coordinate frame. The following assumptions apply [2]:

1. The gravitational acceleration remains constant;

2. Crosswind disturbances and nonlinear aerodynamic terms are neglected;

3. Post-inflation deformations of the parachute are small deformations;

4. The UAV and its attached components are modeled as rigid bodies with defined geometries and masses.

Based on the aforementioned assumptions, and taking into comprehensive consideration generalized inertial forces, active forces, and the effects of flexible deformation of the parachute, the flexible deformation during the airdrop process is approximated using modal superposition, described by a sixth-order vibration equation [29]. The natural frequencies in matrix K are derived from the finite element modal analysis of the parachute canopy, while the damping ratio in matrix C is set to 0.03, a typical empirical value for fabric structures [30].

$$M\ddot{q} + C\dot{q} + Kq = F_{aero} \tag{1}$$

Where: $q = [q_1, q_2, \ldots, q_6]^T$ is the 6th-order modal displacement vector;
$M = \text{diag}(m_1, \ldots, m_6)$ is the modal mass matrix;
$C = \text{diag}(2\zeta_1\omega_{n1}, \ldots, 2\zeta_6\omega_{n6})$ is the diagonal damping matrix;
$K = \text{diag}(\omega_{n1}^2, \ldots, \omega_{n6}^2)$ is the modal stiffness matrix;
$F_{aero} = [F_{aero,1}, \ldots, F_{aero,6}]^T$ is the aerodynamic force vector
Flexible deformation induces variations in structural attitude, where the modal shape coefficient η characterizes the proportional contribution of deformation to the angle of attack:

$$\alpha_{\text{eff}} = \arctan\left(\frac{v_y}{v_x}\right) - \left(\theta + \sum_{i=1}^{6} \eta_i q_i\right) \tag{2}$$

Where: $\alpha_{\text{eff}}$: Effective angle of attack, representing the actual airflow direction relative to the deformed structure's attitude; $v_x, v_y$: horizontal and vertical velocities of the parachute; $\theta$: Parachute attitude angle; $\eta_2 \sim \eta_6$: Shape coefficients of each modal order; $\sum_{i=2}^{6} \eta_i q_i$: Reflects the influence of asymmetric deformation on the angle of attack [30].

Aerodynamic force vector coupling lift and drag formulas:

$$L = \frac{1}{2}\rho V^2 S \left(C_L - \sum_{i=1}^{6} k_{L,i}\dot{q}_i\right) \quad D = \frac{1}{2}\rho V^2 S \left(C_D + \sum_{i=1}^{6} k_{D,i}q_i\right) \tag{3}$$

Where: $\rho$: Air density; $V$: Parachute velocity; $S$: Projected area of the parachute; $k_{D,i}$, $k_{L,i}$: Drag and lift coupling coefficients for each structural mode, obtained by fitting CFD simulation results using the least squares method, representing the sensitivity of drag and lift to the deformation $q_i$ and deformation rate $\dot{q}_i$ of structural mode.

Aerodynamic center offset moment (multi-mode coupling):

$$M = \frac{1}{2}\rho V^2 SL \left(C_{M0} + \sum_{i=1}^{6} k_{M,i}q_i\right) \tag{4}$$

Where: $L$: Parachute diameter; $C_{M0}$: Baseline moment coefficient; $k_{M,i}$: Coupling coefficient of each structural mode to the moment, with the same source as $k_{D,i}$, representing the sensitivity of the moment to the deformation $q_i$ of structural mode. The coordinate systems are defined as follows, as illustrated in Fig 3:

1. Ground Coordinate System ($O_{xyz}$): OX-axis: Aligned with the intersection line of the initial velocity direction plane and the horizontal plane. OY-axis: Directed vertically upward. OZ-axis: Determined by the right-hand rule relative to the OX and OY axis.

2. UAV, Linkage Bearing, and Parachute Body Coordinate Systems ($O_{(1)}X_1Y_1Z_1$, $O_{(2)}X_1Y_1Z_1$, $O_{(3)}X_1Y_1Z_1$): $O_{(1)}$, $O_{(2)}$, $O_{(3)}$ located at the center of mass of each respective component (UAV, linkage bearing, parachute).$X_1$-axis: Coincident with the central axis of the component and pointing in the flight direction. $Y_1$-axis and $Z_1$-axis: Located within the component's transverse plane. These axes rotate with the component.

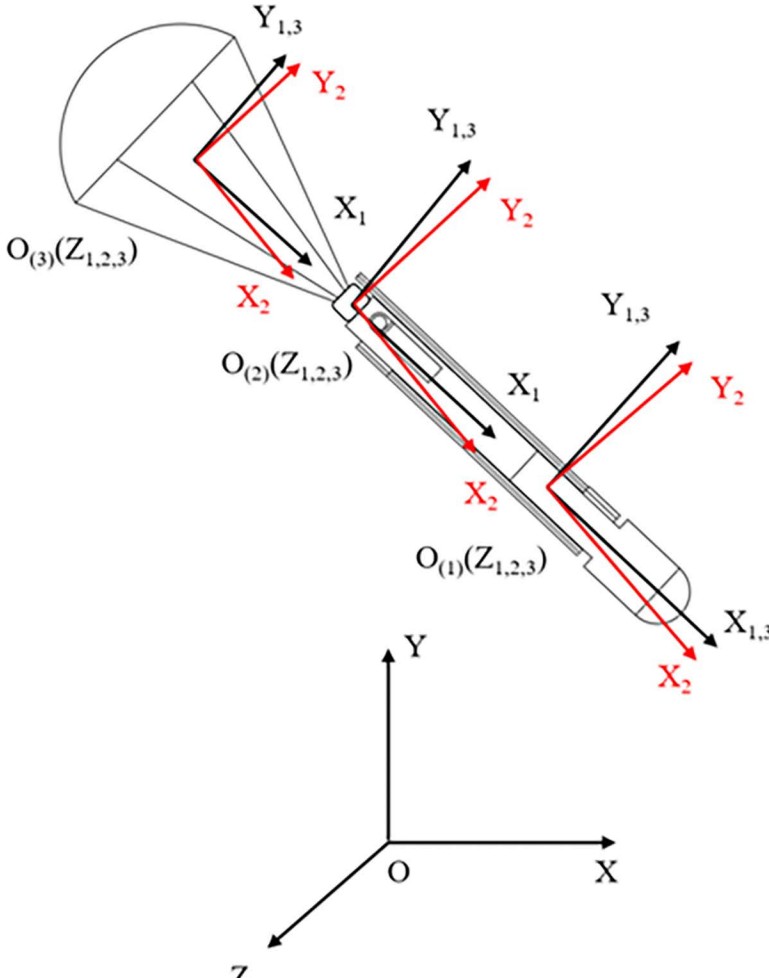

**Fig 3. Parachute-connecting bearing-UAV system coordinate frame.**

3. UAV and Parachute Velocity Coordinate Systems ($O_{(1)}X_2Y_2Z_2$, $O(3)X_2Y_2Z_2$): $O_{(1)}$ for UAV, $O_{(3)}$ for parachute. $X_2$-axis: Aligned with the velocity vector of the component. $Y_2$-axis: Perpendicular to the $X_2$-axis and directed upward within the component's plane of symmetry. $Z_2$-axis: Determined by the right-hand rule relative to the $X_2$ and $Y_2$ axes.

4. UAV and Parachute Non-Rotating Body-Axis Coordinate Systems ($O_{(1)}X_3Y_3Z_3$, $O_{(3)}X_3Y_3Z_3$): $O_{(1)}$ for UAV, $O_{(3)}$ for parachute. $X_3$-axis, $Y_3$-axis, $Z_3$-axis: Aligned with the respective axes ($X_1$, $Y_1$, $Z_1$) of the component's body coordinate system at a specific instant. These axes remain fixed in orientation and do not rotate with the component.

Ground coordinate system serves as an inertial reference frame for describing absolute position vectors and defining the gravitational field.

Body coordinate system fixed to the center of mass of the component; used to define generalized velocities, calculate the partial velocities and partial angular velocities required for Kane's equations, and express inertia properties.

Wind coordinate system aligned with the velocity vector; used to project aerodynamic active forces (lift and drag), which are subsequently transformed into generalized active forces.

Fixed-axis coordinate system aligned with the structural axes but does not rotate with the body; used to describe attitude (Euler angles) and to establish kinematic constraint equations between multi-body components.

**Generalized velocities of the system**

In the dynamics modeling of the parachute – UAV system based on Kane's method, the generalized velocity vector$[u_p]$of the system is composed of motion parameters from three components: the UAV, the linking bearing, and the parachute. The expression is as follows:

$$[u_p] = \begin{bmatrix} \dot{x} & \dot{y} & \dot{z} & \omega_x^{(1)} & \omega_y^{(1)} & \omega_z^{(1)} & \omega_z^{(2,1)} & \omega_x^{(3,2)} & \omega_y^{(3,2)} & \omega_z^{(3,2)} \end{bmatrix}^T \tag{5}$$

the first six terms describe the 6-DOF motion of the UAV, characterizing the translational velocity of the center of mass, as well as the roll, pitch, and yaw angular velocities; $\omega_z^{(2,1)}$ represents the single-DOF revolute motion of the linking bearing relative to the UAV, which is used for roll correction during airdrop operations; the last three terms $\omega_x^{(3,2)} \omega_y^{(3,2)}$ and $\omega_z^{(3,2)}$ describe the three-dimensional attitude adjustment of the parachute relative to the bearing through a spherical joint connection.

**UAV equations of motion**

Kinematic Centroid Vector of UAV:

$$r_c^{(1)} = x\mathbf{i} + y\mathbf{j} + z\mathbf{k} \tag{6}$$

Velocities and Angular Velocities:

$$v_c^{(1)} = \sum_{p=1}^{10} u_p v_p^{(1)} = u_1\mathbf{i} + u_2\mathbf{j} + u_3\mathbf{k} \tag{7}$$

$$\omega^{(1)} = \sum_{p=1}^{10} u_p \omega_p^{(1)} = u_4\mathbf{i}_{x_1} + u_5\mathbf{j}_{y_1} + u_6\mathbf{k}_{z_1} \tag{8}$$

Partial Velocities and Partial Angular Velocities:

$$v_p^{(1)} = i, j, k, \quad \omega_p^{(1)} = 0 \qquad (p = 1, 2, 3)$$

$$v_p^{(1)} = 0, \quad \omega_p^{(1)} = i_{x_1}, j_{y_1}, k_{z_1} \qquad (p = 4, 5, 6) \qquad (9)$$

$$v_p^{(1)} = 0, \quad \omega_p^{(1)} = 0 \qquad (p = 7, 8, 9, 10)$$

## Link bearing dynamics

Link Bearing Position Vector:

$$r_c^{(2)} = r_c^{(1)} + r^{(2,1)} \qquad (10)$$

Velocities and Angular Velocities:

$$v_c^{(2)} = v_c^{(1)} + \omega^{(1)} \times r^{(2,1)} = \sum_{p=1}^{10} u_p v_p^{(2)} \qquad (11)$$

$$\omega^{(2)} = \omega^{(1)} + \omega^{(2,1)} = \omega^{(1)} + u_7 k_{z_1}^{(2)} \qquad (12)$$

Partial Velocities and Partial Angular Velocities:

$$v_p^{(2)} = v_p^{(1)} + \omega_p^{(1)} \times r^{(2,1)} \quad (p = 1, ..., 6)$$

$$v_7^{(2)} = 0, \quad v_p^{(2)} = 0 \quad (p \geq 8) \qquad (13)$$

$$\omega_p^{(2)} = \begin{cases} \omega_p^{(1)}, & p = 1, \ldots, 6 \\ k_{z_1}^{(2)}, & p = 7 \\ 0, & p \geq 8 \end{cases} \qquad (14)$$

## Parachute equations of motion

Parachute Position Vector:

$$r_c^{(3)} = r_{o_3} + r_c^{(3)} \qquad (15)$$

Velocities and Angular Velocities:

$$v_c^{(3)} = v_{o_2} + \omega^{(3)} \times r_c^{(3)} = \sum_{p=1}^{10} u_p v_p^{(3)} \qquad (16)$$

$$\omega^{(3)} = \omega^{(2)} + \omega^{(3,2)} = \omega^{(2)} + \sum_{p=8}^{10} u_p \omega_p^{(3,2)}$$

(17)

$$\omega^{(3,2)} = u_8 i_{x_1}^{(3)} + u_9 j_{y_1}^{(3)} + u_{10} k_{z_1}^{(3)}$$

Partial Velocities and Partial Angular Velocities:

$$v_p^{(3)} = v_p^{(2)} + \omega_p^{(3)} \times r_c^{(3)} \quad (p = 1, \dots, 7)$$

(18)

$$v_p^{(3)} = 0 \quad (p \geq 8)$$

$$\omega_p^{(3)} = \begin{cases} \omega_p^{(2)}, & p = 1, \dots, 7 \\ i_{x_1}^{(3)}, & p = 8 \\ j_{y_1}^{(3)}, & p = 9 \\ k_{z_1}^{(3)}, & p = 10 \end{cases}$$

(19)

## System dynamics equations

Generalized Inertial Forces and Active Forces, Mass Matrix and Damping Matrix (for Components k = 1,2,3):

$$m_{pq}^{(k)} = m^{(k)} v_p^{(k)} \cdot v_q^{(k)} + \omega_p^{(k)} \cdot I^{(k)} \cdot \omega_q^{(k)}$$

(20)

$$c_{pq}^{(k)} = m^{(k)} v_p^{(k)} \cdot \dot{v}_q^{(k)} + \omega_p^{(k)} \cdot \left( I^{(k)} \dot{\omega}_q^{(k)} + \omega_p^{(k)} \times I^{(k)} \omega_q^{(k)} \right)$$

(21)

Generalized Inertial Forces and Active Forces:

$$f_p^{*(k)} = -\sum_{q=1}^{10} \left( m_{pq}^{(k)} \dot{u}_q + c_{pq}^{(k)} u_q \right)$$

(22)

$$f_p^{(k)} = v_p^{(k)} \cdot F^{(k)} + \omega_p^{(k)} \cdot M^{(k)}$$

(23)

Dynamic Equations of the System:

$$\sum_{k=1}^{3} \left( \left[ m_{pq}^{(k)} \right] [\dot{u}_q] + \left[ c_{pq}^{(k)} \right] [u_q] \right) = \sum_{k=1}^{3} \left[ f_p^{(k)} \right]$$

(24)

Where :

$$[m_{pq}] = \sum_{k=1}^{3} m_{pq}^{(k)}, \quad [c_{pq}] = \sum_{k=1}^{3} c_{pq}^{(k)}, \quad [f_p] = \sum_{k=1}^{3} f_p^{(k)}$$

(25)

## FSI simulation analysis

Although the multi-degree-of-freedom multibody dynamics model requires relatively low computational cost, its simplified treatment of external wind fields leads to unclear influence patterns on trajectory prediction. Therefore, validation through both dynamic modeling and bidirectional FSI coupling is required [23]. Bidirectional coupled FSI numerical methodology implements real-time data exchange through integrated solvers: The Computational Fluid Dynamics (CFD) solver computes unsteady aerodynamic loads on the flexible parachute surface and transmits pressure field data to the structural solver. The Computational Structural Dynamics (CSD) solver calculates transient deformation responses using finite element methods, then feeds back displacement boundary conditions to fluid domain for mesh morphing and flow field updates. The computational workflow executes as Fig 4 illustrates.

The computational fluid domain dimensions are 80 meters in length, 30 meters in width, and 150 meters in height (see Fig 5). The blockage ratio is defined as the ratio of the projected area of the fully inflated parachute to the inlet area of the computational domain, which is approximately 0.47% in this setup, below the generally accepted threshold of 3%. Regarding the lateral boundaries, the minimum distance from the parachute center to the nearest sidewall is set to 15 meters, corresponding to approximately 15 times the characteristic diameter, effectively mitigating wall interference effects. Finally, the vertical height is selected to cover a maximum deployment altitude loss of about 120 meters, retaining a downstream

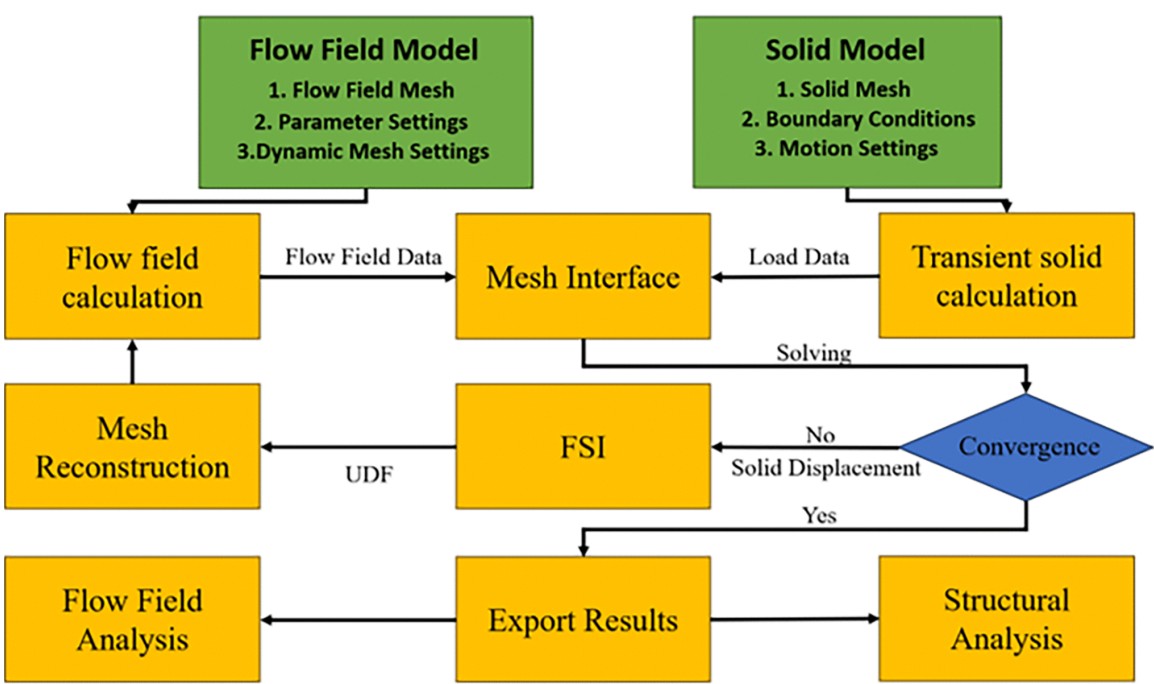

**Fig 4. Flow-solid coupling process.**

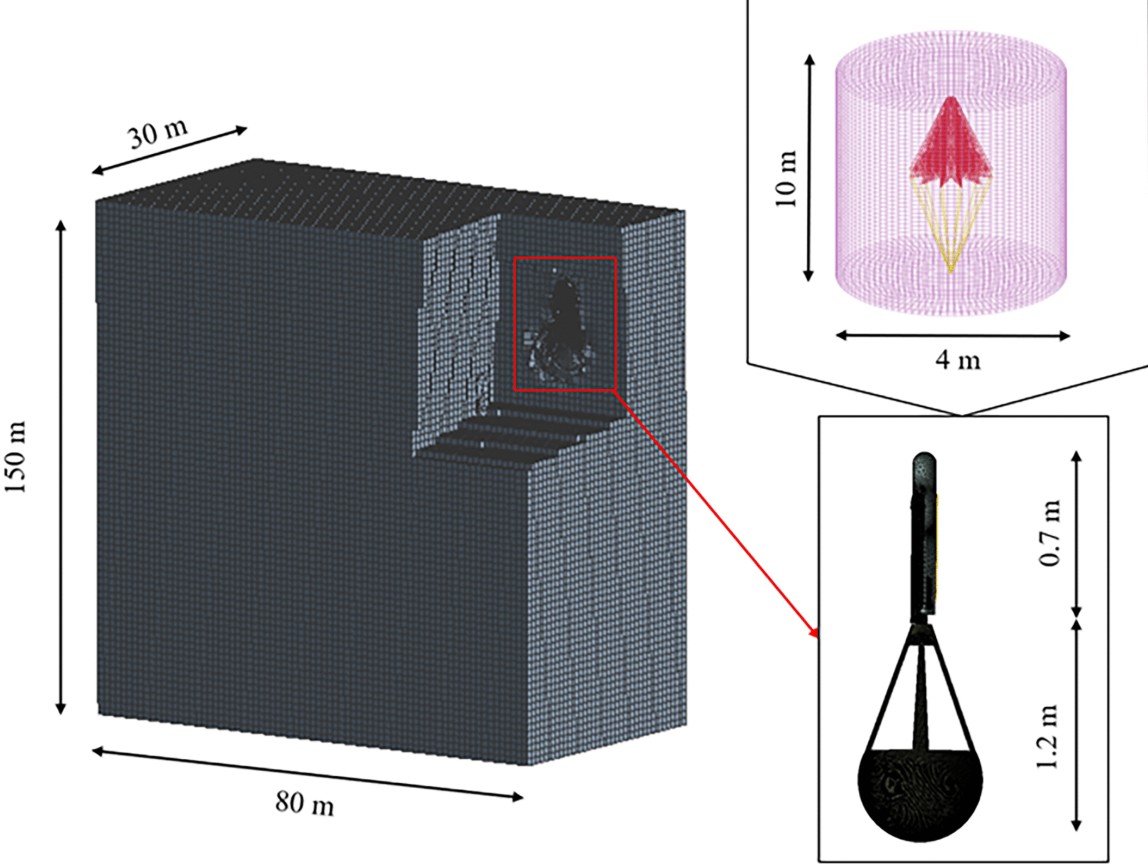

**Fig 5. FSI simulation of flow field and mesh generation.**

buffer zone exceeding 20D, ensuring that the downstream extension region is sufficient for capturing fully developed wake vortex structures without truncation errors.

The mesh system adopts a hexahedral-dominant topology with a no-slip wall boundary condition applied. An inflation layer generation technique is employed, with the first-layer mesh height set to 0.05 mm, a growth rate of 1.2, and a total of 23 layers, achieving a dimensionless wall distance y+ of 10. Key parameters include: high-resolution cells (0.001 m) on the wing surface and parachute riser attachment points; medium-scale meshes (0.004 m) in other regions to balance accuracy and computational efficiency; and adaptive mesh refinement technology dynamically resolving parachute deformation and flow separation phenomena.

Mesh independence verification is essential for the simulation（Table 2）. Taking the case with an initial velocity of 40 m/s and a parachute diameter of 1.0 m as an example: Case 1 uses a high-resolution mesh size of 0.002 m and a medium mesh size of 0.005 m in other regions; Case 2 uses a high-resolution mesh size of 0.001 m and a medium mesh size of 0.004 m in other regions; Case 3 uses a high-resolution mesh size of 0.001 m and a medium mesh size of 0.003 m in other regions. Further mesh refinement led to adaptive mesh refinement failure and computational errors. To ensure computational accuracy while maintaining efficiency, the mesh configuration of Case 2 is ultimately selected.

The horizontal velocity was set to 30 m/s, 35 m/s, 40 m/s, 45 m/s, and 50 m/s based on the carrier aircraft's cruise speed. The vertical velocity was fixed at 10 m/s to represent ejection mechanism dynamics. Parachute diameters of 0.6 m, 0.8 m, 1.0 m, and 1.2 m were selected as key design variables. Velocity and attitude data of the fully inflated

**Table 2. Mesh independence verification.**

| | Number of Mesh | Horizontal Displacement | Vertical Displacement | Horizontal Error | Vertical Error |
|---|---|---|---|---|---|
| 1 | 12215021 | 17.96m | 15.58m | 16.38% | 46.67% |
| 2 | 17574484 | 19.15m | 16.48m | 12.99% | 12.59% |
| 3 | 20159672 | 19.42m | 17.16m | 3.66% | 2.67% |

A numerical simulation of the parachute inflation process was performed using LS-DYNA explicit dynamics analysis. The inflation and deployment process of the parachute was simulated via LS-DYNA explicit dynamics analysis to capture the coupled dynamics of the parachute-riser system. The computational model employed an Arbitrary ALE coupled algorithm, featuring: ALE formulation for the parachute to resolve fluid-structure interaction (FSI); BEAM elements discretizing riser cables with tension-only material properties [26]. Dynamic characteristic parameters of the parachute from deployment to stabilization were obtained through numerical simulation. A two-way implicit coupling strategy was employed to address the interaction between the flexible canopy and airflow. The time step was set to 0.01 s, with a maximum of 30 coupling iterations performed per time step. To accommodate the large deformation of the parachute, adaptive mesh refinement technology was applied to update the fluid domain mesh. The convergence criterion was set to an RMS residual of $10^{-4}$. The results indicate that the parachute fully deploys in approximately 0.8 s. The velocity contours during the deployment process are shown in Fig 6.

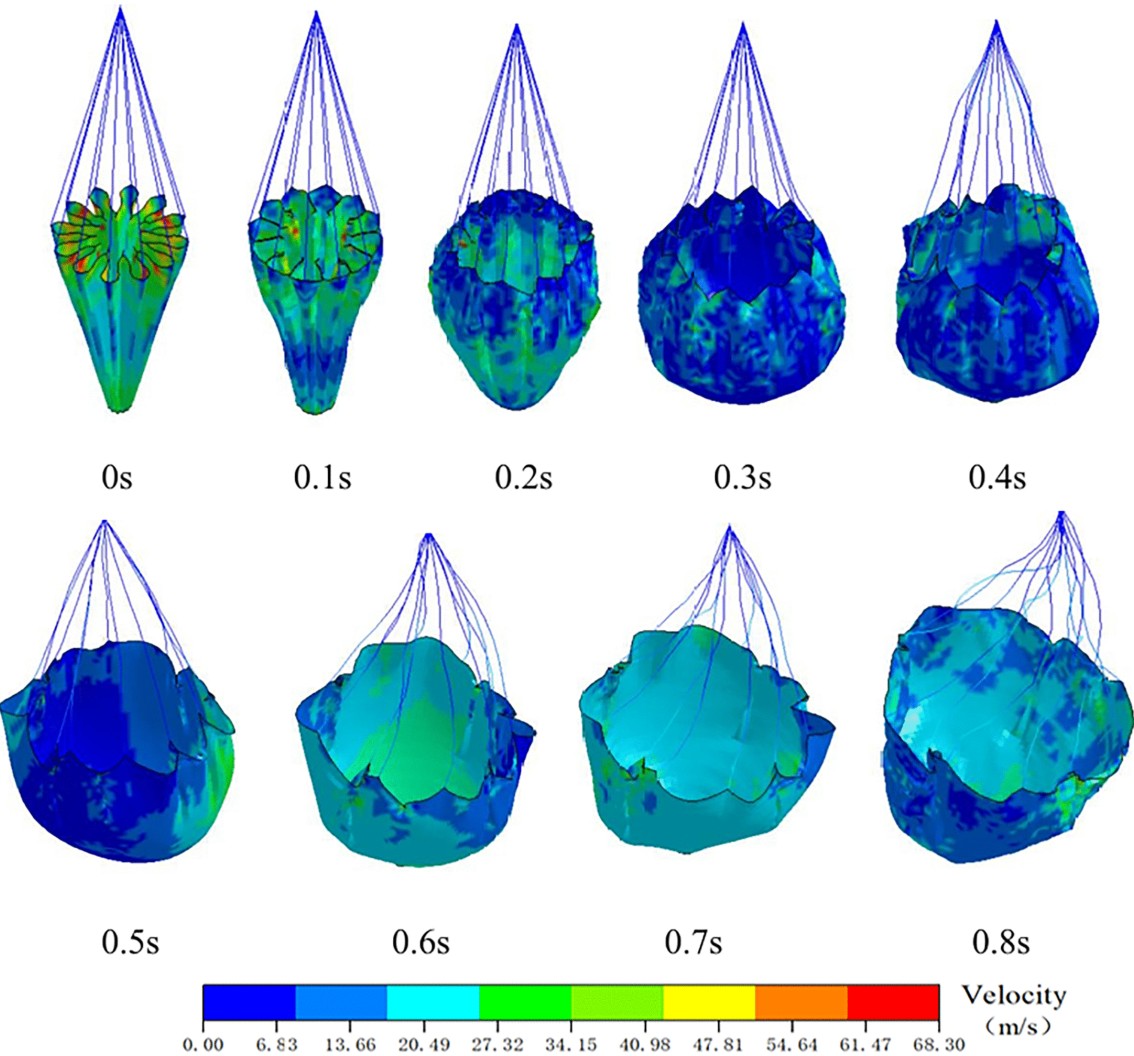

**Fig 6. Velocity field visualization during parachute inflation process.**

parachute, obtained from LS-DYNA simulations, were input into FSI transient analyses. This framework established multivariate parametric combinations to systematically investigate the impact of initial conditions on airdrop trajectories.

A comparative analysis of turbulence models was performed on the airdrop-capable folding-wing UAV system, evaluating the conventional k-ω SST model versus the Gamma-Theta transition model. As demonstrated in Fig 7, the Gamma-Theta transition model achieved smaller deviations in computed lift and drag coefficients from theoretical benchmarks compared to the k-ω SST model.

## Multibody dynamics model validation

On the Matlab platform, a variable-step Runge-Kutta algorithm is employed to numerically solve the ten-degree-of-freedom nonlinear dynamic equations, establishing a high-fidelity numerical simulation environment. By comparing theoretical calculation results with those of the rigid-body model (RBD) and experimental data from CFD fluid-structure interaction simulations, the accuracy of the model in predicting airdrop trajectories is validated (Fig 8 for the trajectory of the parachute-UAV system dynamics model; Fig 9 for the velocity contours of the FSI simulation; Fig 10 for the rigid-body airdrop trajectory comparison; Fig 11 for the FSI airdrop trajectory comparison).

Validation data confirms that both simulation methods achieve coordinate errors within ±5%, mutually validating their efficacy in characterizing the airdrop UAV's dynamics under multiparameter coupling effects (Table 3). The parachute diameter dominates system behavior: reducing it from 1.2 m to 0.6 m triggers a 320%–410% nonlinear enhancement in horizontal displacement capability. In contrast, increasing initial velocity from 30 m/s to 50 m/s yields only 38.7±2.5% improvement in horizontal displacement. Vertically, parachute diameter exhibits exponential scaling with descent distance—systems with 1.2 m canopies stabilize at −15.2±0.8 m, while 0.6 m counterparts require −59.3±2.7 m for stabilization. Attitude dynamics analysis reveals 40% longer stabilization times and significantly extended settling distances at 50 m/s versus low-speed cases. Notably, under high-speed (≥45 m/s) and small-parachute (<0.8 m) conditions, FSI simulations deviate substantially from predicted patterns, showing >15% horizontal position error and >30% vertical distance discrepancy, necessitating additional deviation analysis.

We conducted an airdrop flight test under the conditions of an initial velocity of 40 m/s and a parachute diameter of 1.2 m. The test site is illustrated in the Fig 12, captured from the belly of the mother aircraft. A full-scale drone prototype and its matching parachute were used for the field airdrop test. During the descent, real-time motion data were synchronously

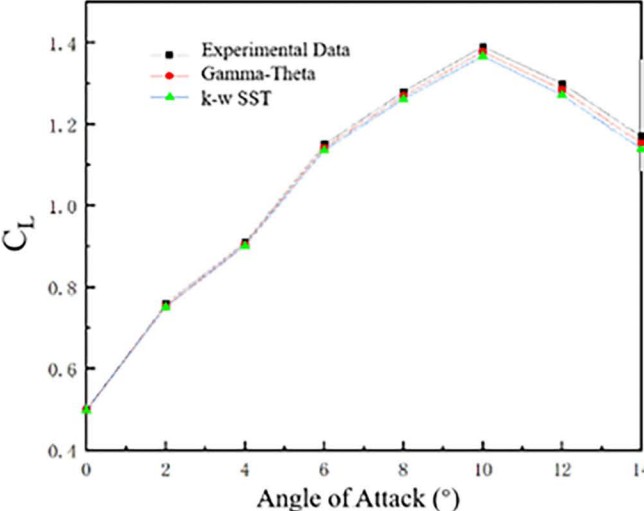 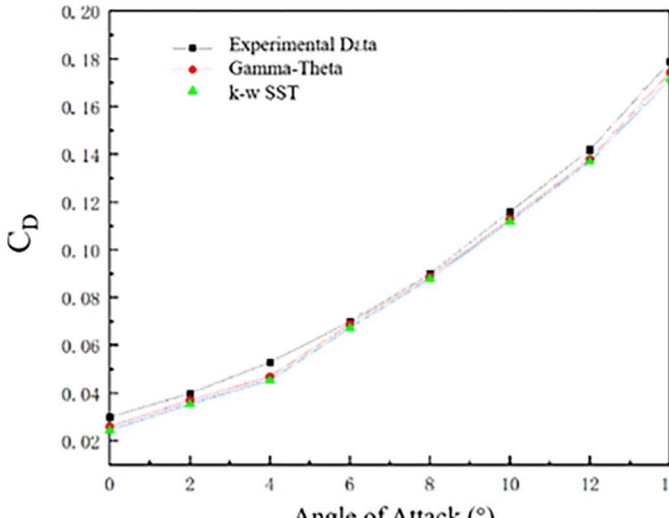

**Fig 7. Comparison curves of lift coefficient and drag coefficient.**

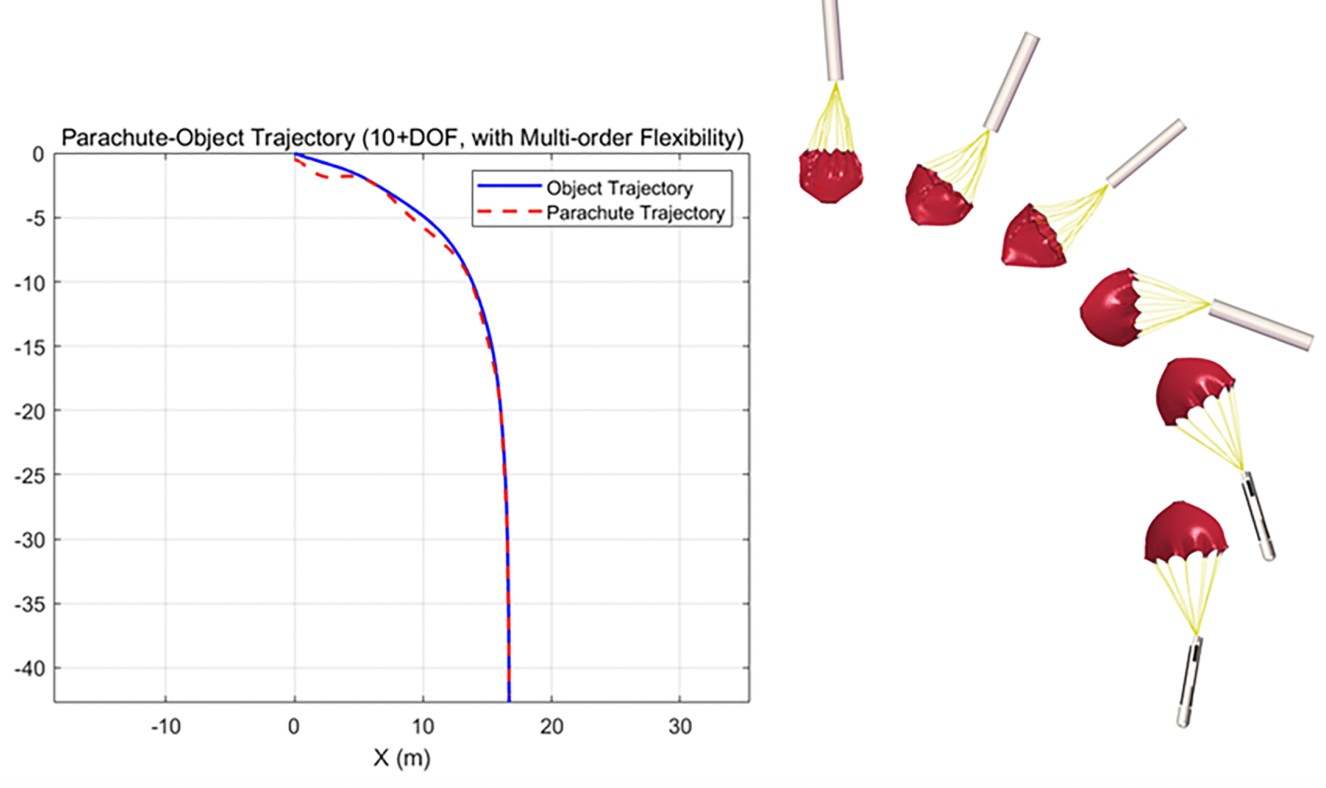

**Fig 8. Parameter diagram considering flexible deformation.** (1.0m parachute diameter and 40m/s starting speed.).

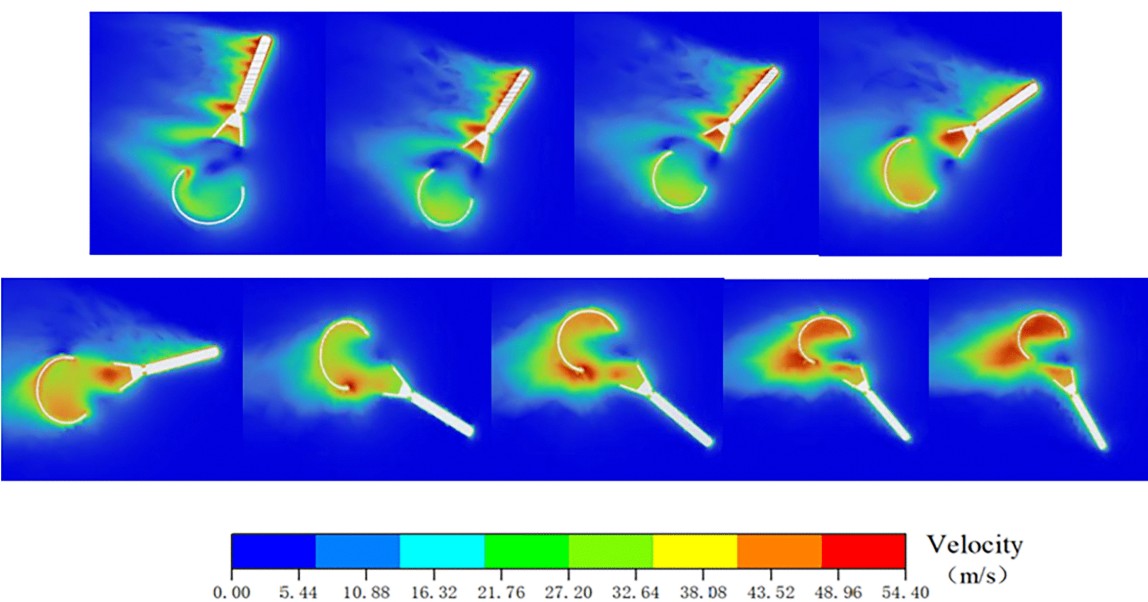

**Fig 9. Velocity contour simulation of airdrop process.**

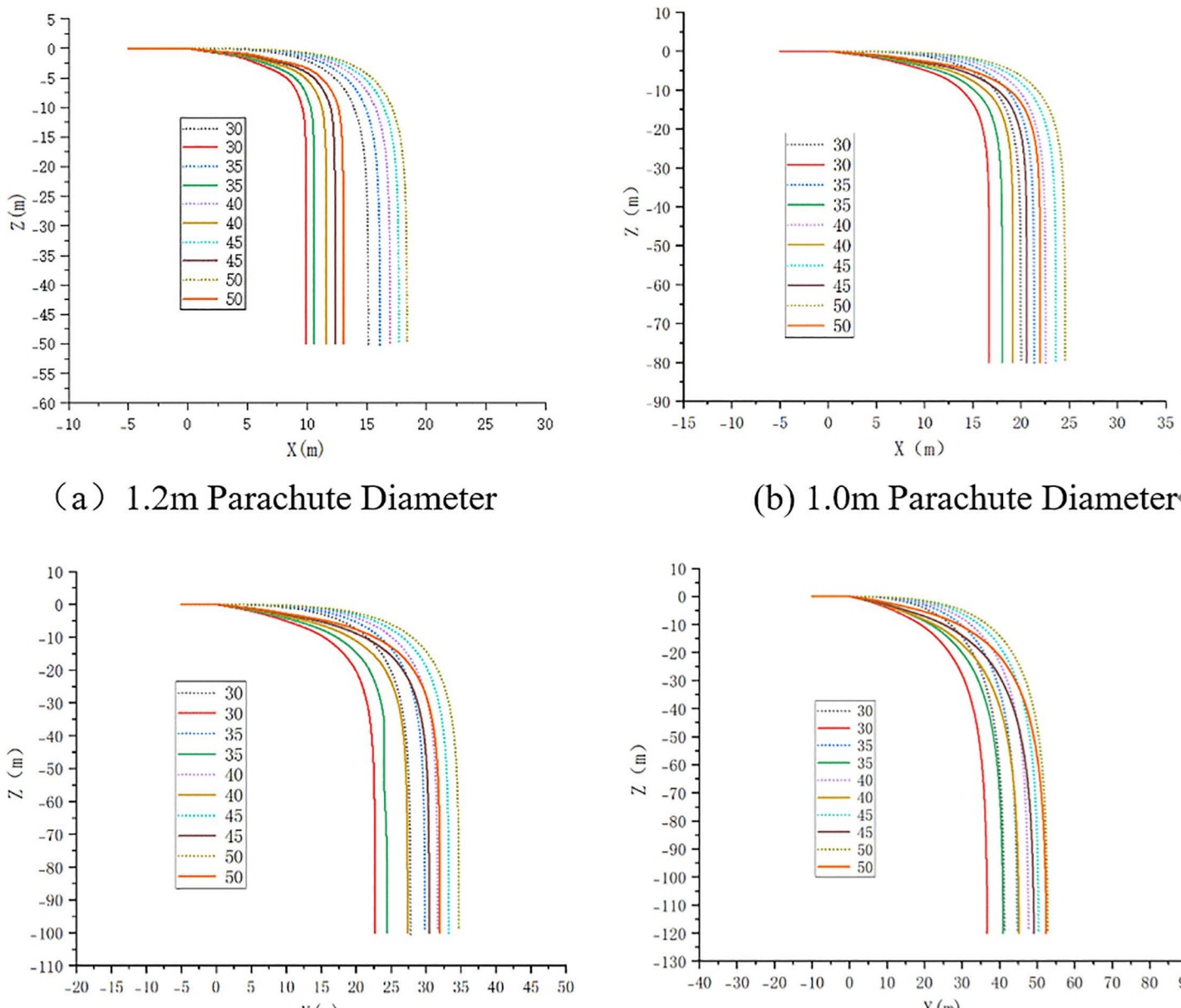

（a）1.2m Parachute Diameter

(b) 1.0m Parachute Diameter

（c）0.8m Parachute Diameter

(d) 0.6m Parachute Diameter

**Fig 10. Comparison of UAV rigid-body trajectories.** (Solid line: multibody dynamics model; Dashed line: rigid-body model.).

recorded by onboard high-precision IMU and GPS sensors. The Fig 13 and Table 4 compare the trajectories of three simulation models with the flight test trajectory, and the corresponding trajectory errors are listed in the table. The results show that the Fluid-Structure Interaction (FSI) model exhibits the closest agreement with the flight test trajectory. The multibody dynamics model maintains relatively low error while significantly reducing computational cost compared to the FSI simulation. In contrast, the Rigid Body Dynamics (RBD) model shows a comparatively larger deviation.

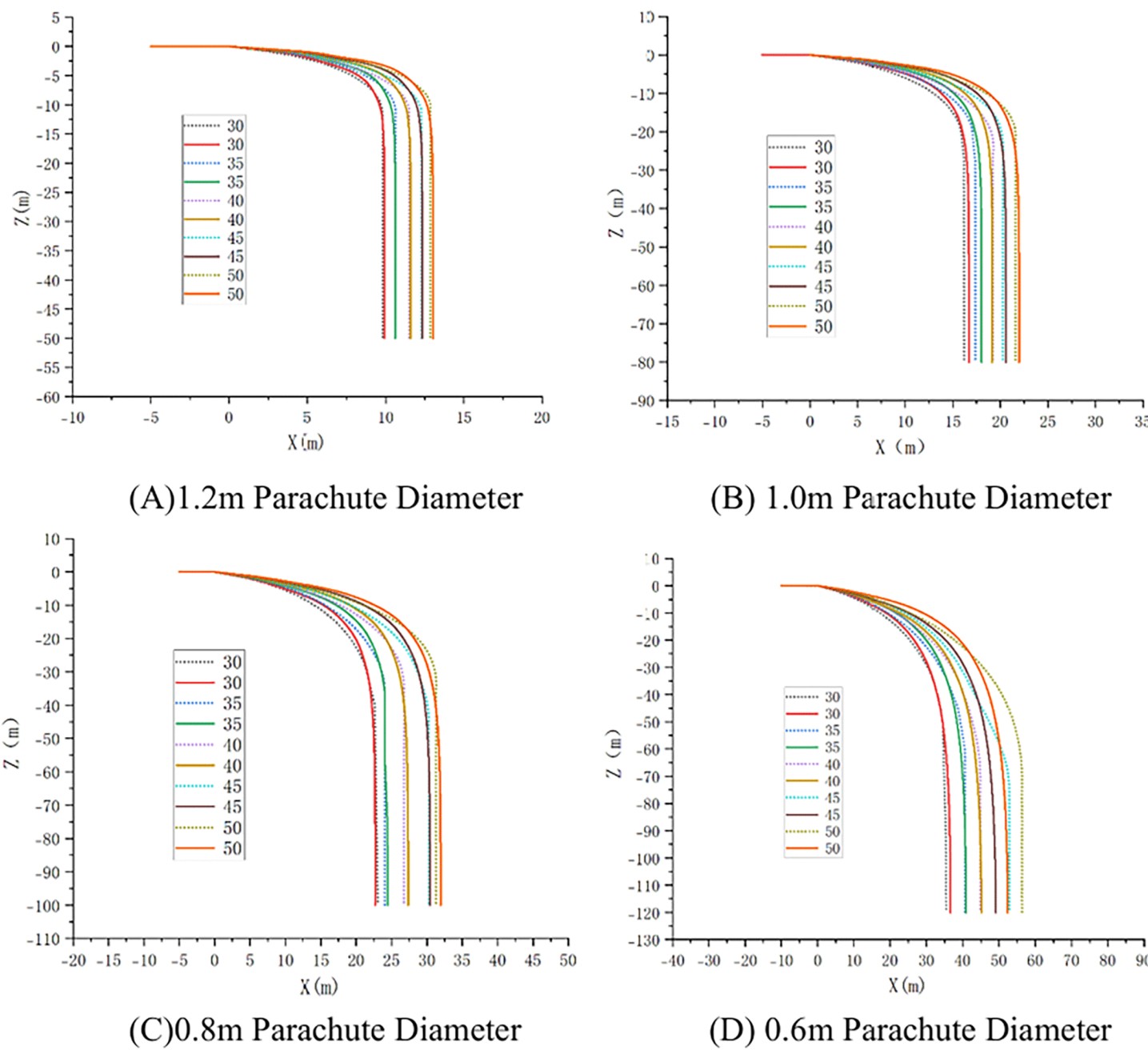

**Fig 11. Comparison of UAV FSI trajectories.** (Solid line: multibody dynamics model; Dashed line: FSI simulation.).

## Airdrop trajectory deviation analysis

The operating conditions cover an angle of attack range of 90°±30° and an initial velocity range of 0–50 m/s. To enhance payload survivability and minimize exposure time within the wake interference region, the initial deployment velocity should be less than or equal to the upper limit of the system design envelope, approximately 45 m/s. Concurrently, as

**Table 3. Comparison table of trajectory errors for different models.**

|  | Error | 0.6m | 0.8m | 1.0m | 1.2m |
|---|---|---|---|---|---|
| RBD | Horizontal Error | 6.99% | 15.30% | 16.38% | 46.67% |
| RBD | Vertical Error | 32.94% | 16.73% | 12.99% | 12.59% |
| FSI | Horizontal Error | 4.40% | 4.38% | 3.66% | 2.67% |
| FSI | Vertical Error | 7.79% | 4.90% | 3.66% | 2.67% |

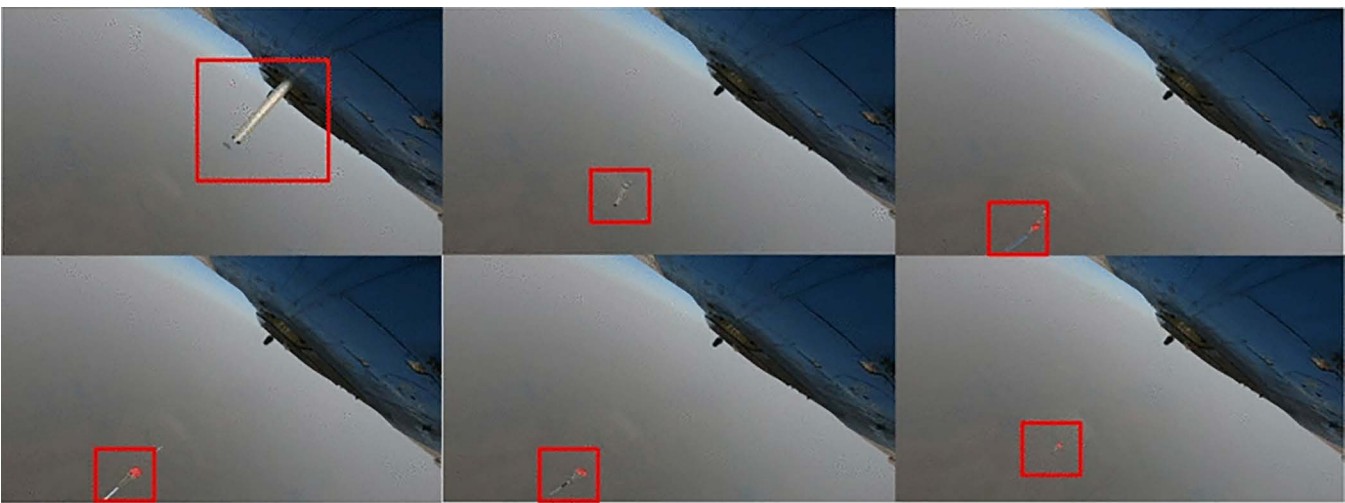

**Fig 12. Aerial view of air drop test site (Mother Ship Perspective).**

shown in Fig 14, when the velocity exceeds 45 m/s and the parachute diameter is less than 0.8 m, the aerodynamic destabilization effect becomes significantly amplified, and the trajectory deviation exhibits a pronounced nonlinear growth trend. Therefore, the condition with an initial velocity ≥ 45 m/s and a parachute diameter < 0.8 m is defined as an extreme operating scenario, reflecting the practical requirements of rapid deployment tasks in complex environments and representing the critical threshold for system stability.

Under this operating condition, Fig 14 reveals the formation of a localized high-pressure zone on the windward side of the canopy and a large-scale low-pressure belt on the leeward side. The impact of the high-speed flow field induces significant nonlinear deformation of the parachute, leading to dynamic distortion of the aerodynamic shape [24]. We quantitatively compared the aerodynamic outputs of the fluid-structure interaction model with those of the rigid-body reference. As shown in the pressure contours in Fig 14, flow-induced deformation causes a notable redistribution of the surface pressure load. Specifically, the deformation results in a rearward shift of the aerodynamic center of the canopy by approximately 26% compared to the rigid-body model. This dynamic pressure differential triggers a degradation in structural geometric stiffness—reduced bending stiffness in the smaller canopy diameter leads to wrinkling deformation at the canopy edge [25]—while simultaneously inducing three major effects: 1. flow separation around the canopy; 2. rearward shift of the aerodynamic center; and 3. destabilizing pitching moment [27]. Streamline observations indicate that the pronounced circumferential spillover effect forms a self-excited oscillation loop: pressure gradients intensify structural deformation → deformation amplifies fluid spillover → energy dissipation imbalance ultimately results in trajectory deviations exceeding the predicted value by >30%.

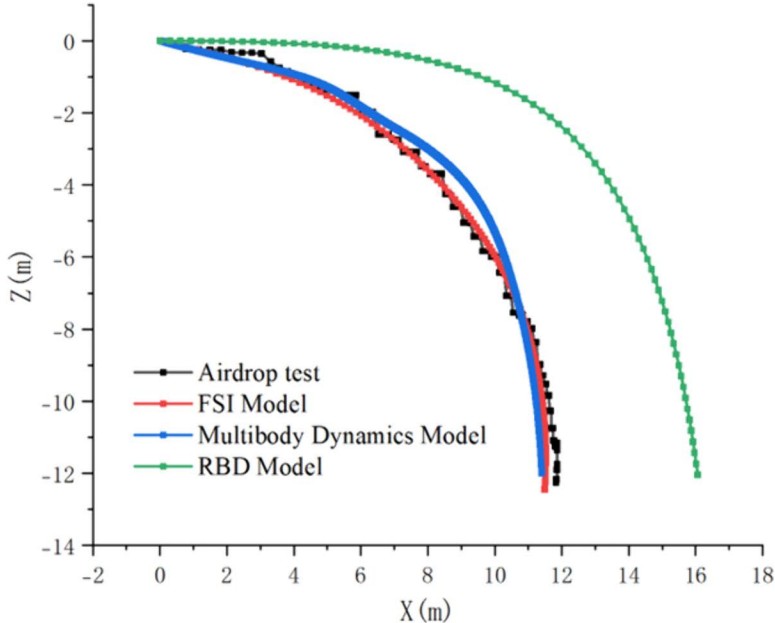

**Fig 13. Comparison of air drop test trajectories.**

**Table 4. Comparison Table of Model and Flight Test Trajectory Errors.**

| Modle | Horizontal Error | Vertical Error |
|---|---|---|
| Multibody Dynamics Model | 3.63% | 7.53% |
| FSI Model | 2.87% | 6.34% |
| RBD Model | 35.24% | 32.7% |

## Parachute jettison points selection

The selection of the parachute release point primarily considers the pitch angle and three-degree-of-freedom directional velocities. The constraint on the horizontal velocity component (Vx < 3 m/s) helps mitigate nonlinear effects in lateral dynamics post-separation, ensuring controller robustness under parametric uncertainties. The lower limit for vertical velocity (Vz ≥ 10 m/s) prevents the peak load from exceeding the allowable structural stress due to an excessively short parachute deployment impact time Δt. Meanwhile, the upper limit (Vz ≤ 20 m/s) is constrained by trajectory control accuracy. The pitch angle convergence threshold (θ < 5°) ensures the system remains within the stable domain of the short-period mode, effectively suppressing pitch oscillations after separation and avoiding divergent motion induced by aerodynamic coupling [10]. For different platforms and external environments, these boundaries must be reevaluated based on specific structural limits, aerodynamic derivatives, and mission requirements.

Implementing a pitch angle convergence threshold (θ < 5°) guarantees the system operates within the short-period mode stability region. This effectively suppresses post-separation pitch oscillations and prevents aerodynamic coupling-induced divergent motion.

With reference to existing constraints, random perturbation values for each deviation factor were incorporated into the ideal trajectory, and Monte Carlo simulations of the system's impact point after parachute release were conducted, as illustrated in Fig 15. The velocity and angle thresholds for the parachute release point are determined under the

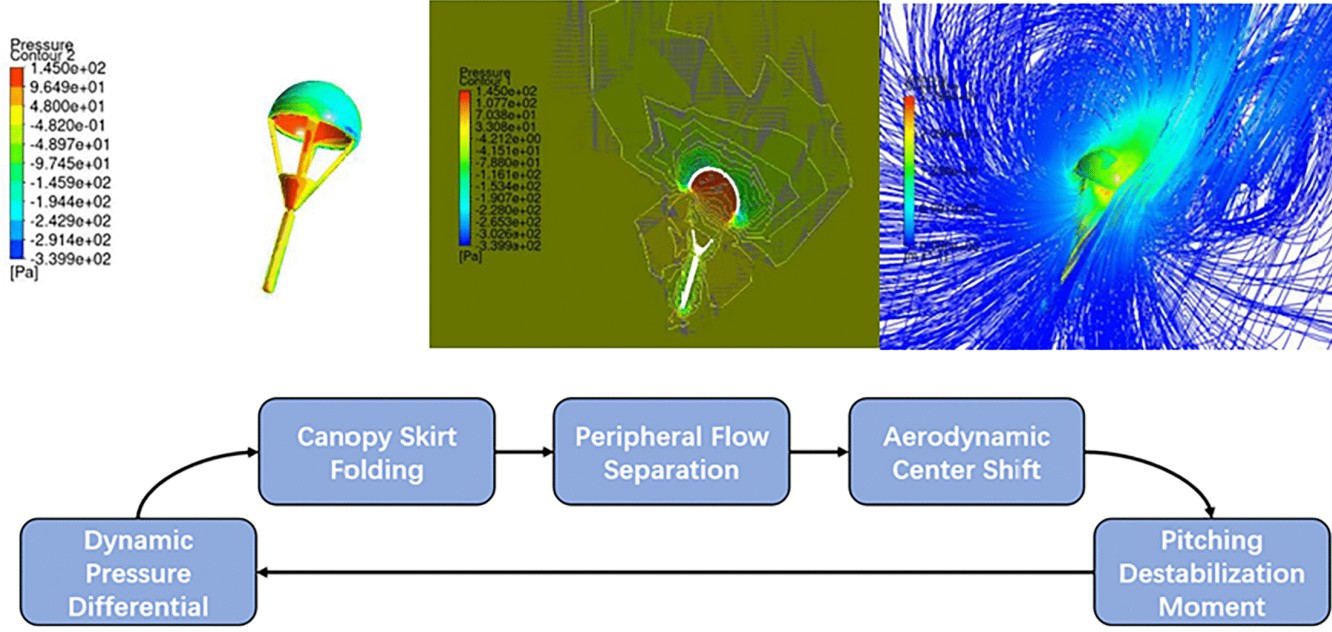

**Fig 14. Airdrop Trajectory Deviation Analysis.** (initial speeds of 45 m/s and 50 m/s.).

combined influence of engineering constraints. For a given deployment altitude, the horizontal velocity threshold ensures that the impact point tolerance remains within the task accuracy requirements, while also helping to reduce the risk of secondary collisions or entanglement between the parachute and the payload after release. Meanwhile, the vertical velocity and pitch angle thresholds create favorable conditions for subsequent system attitude adjustment and stable pull-up maneuvers.

Results demonstrate that when simultaneously satisfying all constraints: Pitch angle convergence $|\theta| < 5$; Horizontal velocity $Vx < 3m/s$; Lateral velocity $-1m/s \leq V_y \leq 1m/s$; Vertical velocity $10m/s \leq V_z \leq 20m/s$. Landing accuracy achieves $< 10m$ deviation with >95% mission success probability, optimally balancing impact load mitigation and trajectory control precision.

The trajectory and parachute jettison points are presented in Fig 16, with the corresponding coordinates detailed in Table 5. Research demonstrates that under constant parachute diameter conditions, horizontal initial velocity exhibits weak sensitivity on cutting point coordinates. This normalized sensitivity coefficient further attenuates below 5% for large parachute (D ≥ 1.0 m) configurations, resulting from the high-damping characteristics of large-diameter systems significantly suppressing initial kinetic energy disturbances. When initial velocity remains constant, variation in parachute diameter exerts a substantially stronger influence on the longitudinal coordinate of the cutting point than on the lateral coordinate ($|\partial y_c/\partial D|/|\partial x_c/\partial D| \approx 1.6$)). This differential effect decays following a negative exponential trend with increasing parachute diameter.

## Conclusions

This study establishes a rigid-flexible coupled dynamics framework for airdrop-capable UAV systems based on Kane's equations, developing a 10-degree-of-freedom model integrating rigid-body motion with parachute deformation characteristics. High-fidelity co-simulation combining LS-DYNA explicit dynamics and Gamma-Theta transition model-based CFD was employed to simulate the entire airdrop process, achieving cross-validation with theoretical trajectories. Results

 

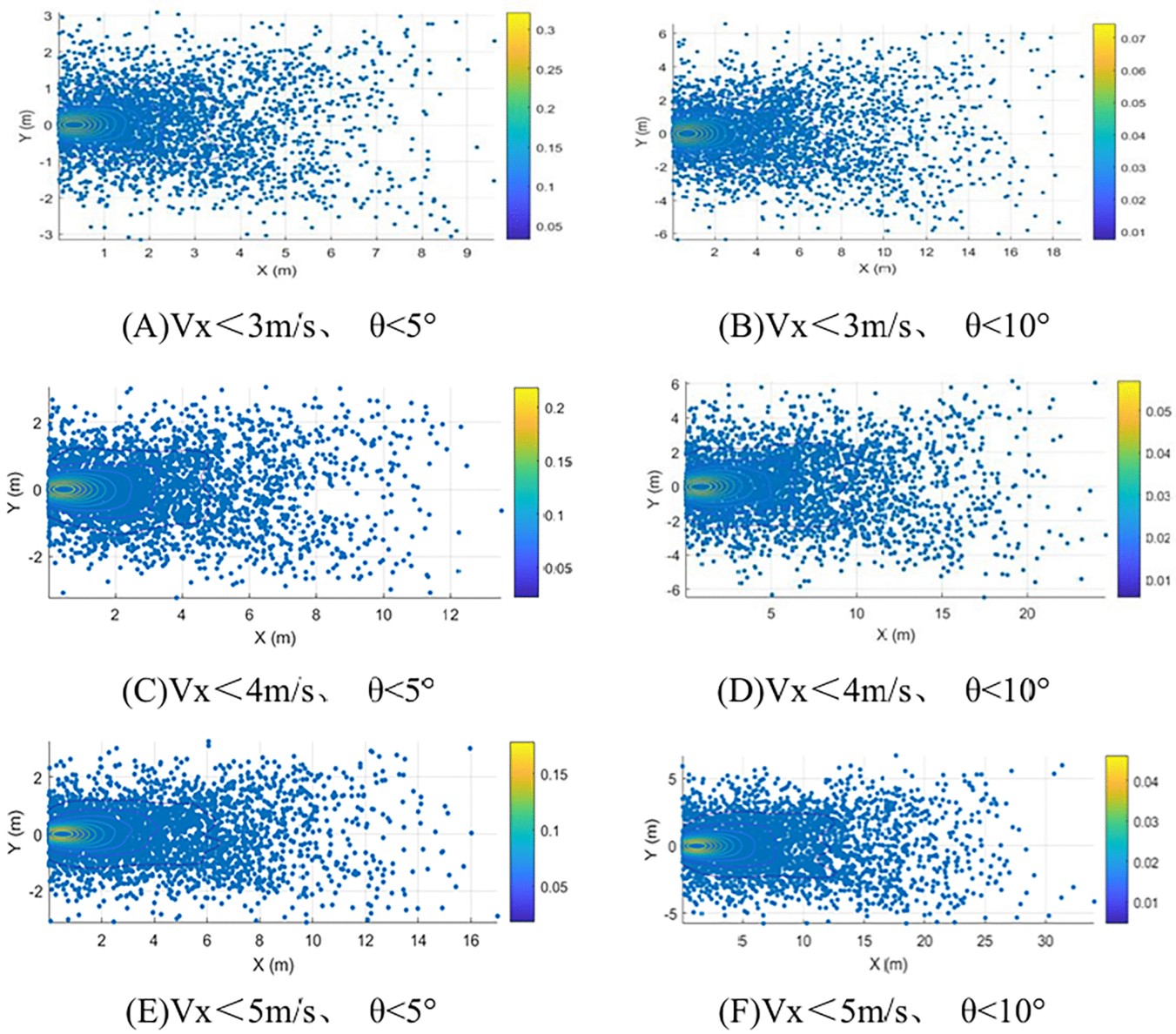

**Fig 15. Selection of parachute jettison timing for parachute – UAV systems.**

demonstrate ≤5% relative error between Kane-derived dynamics and co-simulation results. The work reveals coupling mechanisms between initial horizontal velocity and parachute aerodynamic parameters that govern trajectory dynamics and optimal separation point selection, providing novel insights into multi-physics modeling for complex airdrop systems and validating its engineering efficacy.

The rationality of the model assumptions in this study is primarily based on the following: within the operational altitude variation of approximately 500 meters, the change in gravitational acceleration is less than 0.04%, making the impact of the constant gravity assumption on trajectory prediction negligible; this work focuses on the inherent stability of the system, with environmental disturbances to be analyzed in subsequent robust control studies; the fluid-structure interaction comparison in Section [X]

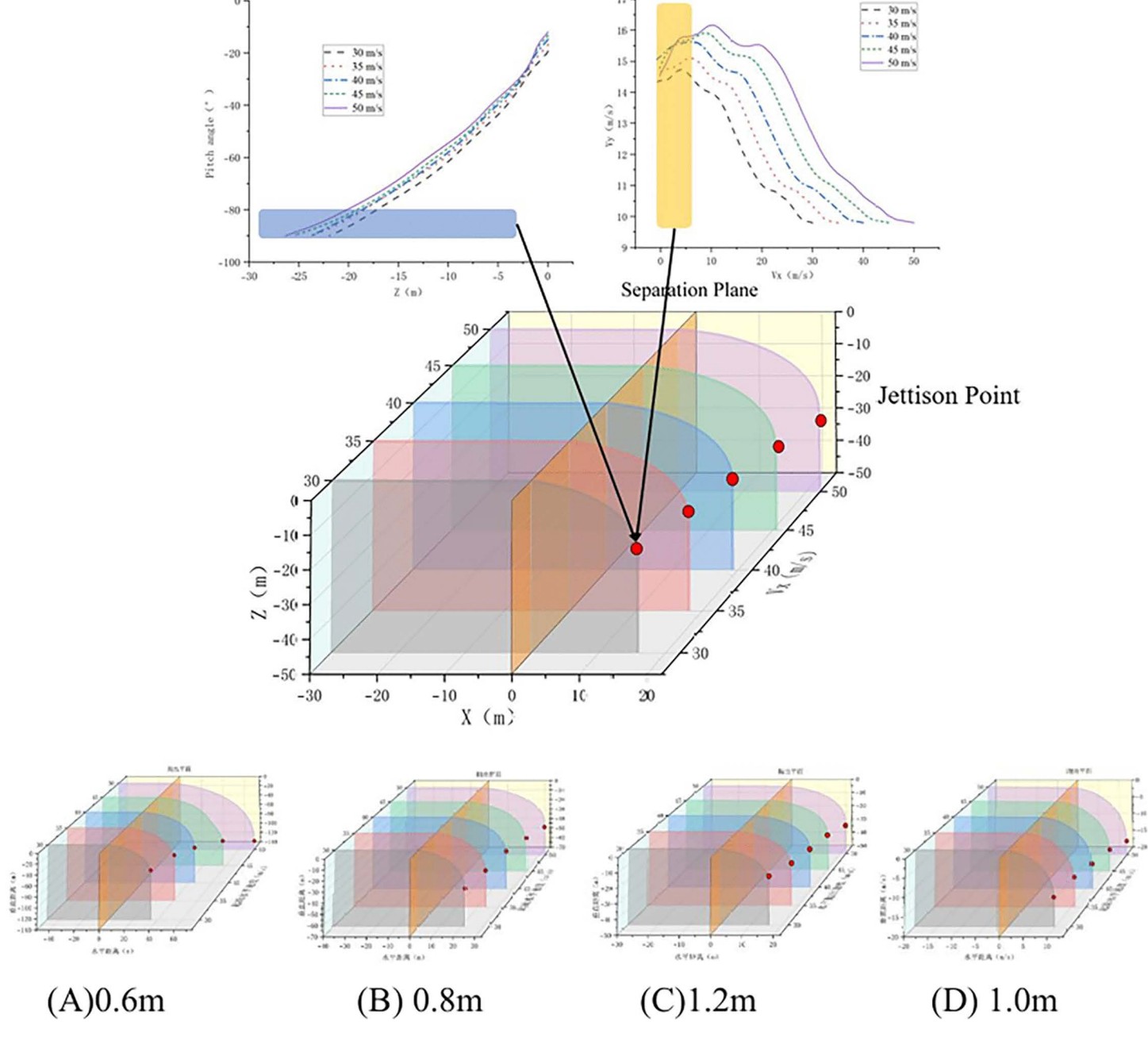

**Fig 16. Parachute size-dependent airdrop trajectories and jettison point.**

**Table 5. Jettison point coordinates under different operating conditions.**

|  | 30m/s | 35m/s | 40m/s | 45m/s | 50m/s |
|---|---|---|---|---|---|
| 0.6m | (34.2, −48.1) | (40.5, −52.7) | (44.6, −65.5) | (54.7, −73.7) | (73.8, −107.2) |
| 0.8m | (21.6, −31.9) | (23.9, −32.8) | (26.7, −33.1) | (30.7, −34.3) | (32.9, −35.3) |
| 1.0m | (15.9, −20.3) | (17.9, −21.8) | (19.2, −22.2) | (20.5, −23.4) | (22.1, −24.2) |
| 1.2m | (9.8, −12.2) | (10.8, −13.0) | (11.5, −13.4) | (12.2, −13.7) | (13.2, −15.0) |

indicates that the trajectory error between the simplified model and the high-fidelity model is less than 5%, demonstrating that the "small deformation" assumption is sufficient to capture the dominant dynamic characteristics during the stable descent phase.

The limitations of this study are as follows: the established dynamic model incorporates necessary simplifications in environmental modeling, failing to fully account for the impact of meteorological conditions such as wind disturbances, wind shear, and atmospheric turbulence on the system's dynamic behavior; additionally, the current model does not cover the full-process dynamic coupling analysis of "launch from tube—parachute descent—parachute release—pull-up and climb," which restricts in-depth exploration of the UAV's integrated control strategies during critical transition phases.

To address the aforementioned limitations, future research will focus on the following two aspects: developing high-fidelity wind field and turbulence models to enhance the reliability and environmental adaptability of flight stability analysis; constructing a refined dynamic model that covers all phases, including launch, descent, parachute deployment, and climb, to systematically study the multi-phase coupled dynamic behavior. Based on this, robust control strategies adapted to the entire flight process will be developed, providing theoretical foundations and methodological support for the engineering application of UAVs in complex mission scenarios.

## Supporting information

**S1 Data. UAV Test Flight, Fluid–Structure Interaction and Dynamics Simulation Data.**
(ZIP)

## Author contributions

**Conceptualization:** Hanxu Guo, Ziang Gao, Jian Zhang.

**Data curation:** Hanxu Guo, Ziang Gao, Miao Zhang.

**Formal analysis:** Hanxu Guo, Ziang Gao.

**Funding acquisition:** Zijian Zhu, Jian Zhang.

**Investigation:** Hanxu Guo, Zijian Zhu, Miao Zhang, Jian Zhang.

**Methodology:** Hanxu Guo, Miao Zhang.

**Project administration:** Jian Zhang.

**Resources:** Zijian Zhu, Miao Zhang.

**Software:** Hanxu Guo, Zijian Zhu.

**Supervision:** Ziang Gao, Jian Zhang.

**Validation:** Hanxu Guo, Ziang Gao.

**Writing – original draft:** Hanxu Guo.

**Writing – review & editing:** Hanxu Guo, Ziang Gao, Jian Zhang.

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
