## [Decision Letter · Decision Letter 0]

7 Dec 2025

Dear Dr. Zhang,

Thank you for submitting your manuscript to PLOS ONE. After careful consideration, we feel that it has merit but does not fully meet PLOS ONE’s publication criteria as it currently stands. Therefore, we invite you to submit a revised version of the manuscript that addresses the points raised during the review process.

We look forward to receiving your revised manuscript.

Kind regards,

Pan Yu

Academic Editor

PLOS One

Journal Requirements:

4. In the online submission form you indicate that your data is not available for proprietary reasons and have provided a contact point for accessing this data. Please note that your current contact point is a co-author on this manuscript. According to our Data Policy, the contact point must not be an author on the manuscript and must be an institutional contact, ideally not an individual. Please revise your data statement to a non-author institutional point of contact, such as a data access or ethics committee, and send this to us via return email. Please also include contact information for the third party organization, and please include the full citation of where the data can be found.

Additional Editor Comments:

A study on parachute-UAV airdrop systems is presented in this manuscript, for which four reviews have been gained. While the methodology is promising and the analysis is systematic, all reviewers have raised substantial concerns that must be addressed for the work to be considered for publication. The authors are strongly encouraged to provide a detailed point-by-point response.

Reviewers' comments:

Reviewer's Responses to Questions

**Comments to the Author**

1. Is the manuscript technically sound, and do the data support the conclusions?

Reviewer #1: No

Reviewer #2: Yes

Reviewer #3: Yes

Reviewer #4: Yes

2. Has the statistical analysis been performed appropriately and rigorously?

Reviewer #1: No

Reviewer #2: Yes

Reviewer #3: Yes

Reviewer #4: Yes

3. Have the authors made all data underlying the findings in their manuscript fully available?

Reviewer #1: Yes

Reviewer #2: Yes

Reviewer #3: No

Reviewer #4: Yes

4. Is the manuscript presented in an intelligible fashion and written in standard English?

Reviewer #1: Yes

Reviewer #2: Yes

Reviewer #3: Yes

Reviewer #4: Yes

Reviewer #1: This study focuses on the parachute recovery system for airdrop-capable unmanned aerial vehicles (UAVs), proposing an innovative research methodology that integrates a 10-degree-of-freedom (10-DOF) multi-body dynamics model based on Kane's equations with a high-fidelity fluid-structure interaction (FSI) co-simulation. Through parametric analyses (e.g., parachute diameter, initial velocity), it reveals trajectory control laws and proposes an optimal parachute jettison point selection strategy based on dynamic constraints. The research emphasizes demonstrating simulation reliability via cross-validation (showing ≤5% error between the dynamics model and FSI results), aiming to provide a theoretical framework for precise UAV airdrops. Based on core academic standards, this manuscript is recommended for rejection due to the lack of physical experimental validation, unquantified model simplification assumptions, and absence of key methodological details.The primary reasons for recommending rejection are as follows:

1. Although the paper employs complex cross-validation of simulations, all results stem from "simulation-to-simulation" comparisons, lacking final verification against real-world physical experimental data (e.g., high-speed photogrammetric trajectory measurements, sensor-based attitude/velocity data). In engineering research, simulation models must be validated against experimental data to verify their boundary conditions and actual accuracy; otherwise, their effectiveness in real-world environments cannot be proven .

2. The paper overlooks key environmental factors (e.g., external wind field disturbances, nonlinear aerodynamic effects) and fails to quantify the impact degree of these simplifications through sensitivity analysis. For instance, ignoring wind fields under complex atmospheric turbulence could lead to trajectory prediction deviations exceeding 20%, but the authors do not discuss the range of such errors, undermining the model's practical applicability .

3. The method for obtaining aerodynamic coefficients (e.g., lift/drag coefficients) in the multi-body dynamics model is not detailed, merely mentioned as "based on CFD data" without specifying the calibration process or citing sources. Furthermore, the absence of key parameters in the FSI simulation, such as grid convergence analysis and time step settings, prevents other researchers from replicating the experiment .

4. The literature review merely lists existing methods without critically pointing out their limitations (e.g., most models do not couple time-varying wind fields with multi-temperature zone cooperative optimization). An analysis of the research gap should be added to clarify this paper's innovation points .

5. The calibration method for aerodynamic coefficients is ambiguous, stated only as "based on CFD data" without specifying the specific conditions (e.g., angle of attack range, Reynolds number). It is suggested to present comparative errors between calibration data and experimental values in a table.

6. Comparisons with baseline algorithms are mentioned as "superior" without presenting specific data (e.g., convergence iteration count, CPU time). Quantitative metrics need to be supplemented .

Reviewer #2: 1 Check the grammar of the entire manuscript. It needs to be further made more accurate to improve readability.

2 Introduction, “Current mainstream air transport methods... low transport efficiency [1].”…….., This paragraph only points out the shortcomings of traditional air transport methods but fails to clarify the specific deficiencies of existing airdrop UAVs in parachute-UAV coupling modeling (e.g., traditional models ignore the impact of parachute flexible deformation on trajectory, insufficient integration of FSI simulation and multibody dynamics). It is recommended to supplement the direct correlation between existing research gaps and the work of this paper to strengthen the research motivation.

3 Research object, “Control is exclusively implemented via two control surfaces positioned at the left and right trailing edges of the rear wing.”

Key parameters of the control surfaces (e.g., deflection angle range, response time, control efficiency coefficient) are not explained. These parameters are crucial for the attitude control of the UAV after separation. It is recommended to supplement relevant parameters to improve the model description of the research object.

4 Table 1 suggests consistent units, mm/m

5 FSI simulation analysis, “The computational fluid domain measuring 80 m (L) × 30 m (W) × 150 m (H) is illustrated in Fig 5.”

The basis for selecting the computational fluid domain size (80m×30m×150m) is not explained, such as whether the maximum deployment size of the parachute and flow field boundary effects (avoiding wall interference) are considered. It is recommended to supplement the rationality analysis for size determination

6 Multibody dynamics model validation, “Validation data confirms that both simulation methods achieve coordinate errors within ±5%”, The statistical type of error (e.g., Root Mean Square Error RMSE, Maximum Absolute Error MAE, average relative error) and sample size (e.g., statistical results of how many time nodes or spatial coordinate points) are not clarified. It is recommended to supplement the specific method and statistical details of error calculation to improve result credibility.

7 Airdrop trajectory deviation analysis, “under extreme operating conditions characterized by initial velocities ≥45 m/s and parachute diameters <0.8 m”, The basis for determining "extreme operating conditions" is not defined (e.g., load limits based on engineering application scenarios, operating condition classification standards in similar studies in the literature, or critical conditions where trajectory deviation increases significantly in simulations). It is recommended to supplement the definition basis to make the operating condition classification more reasonable.

8 Parachute jettison points selection, “Limiting the horizontal velocity component (Vₓ<3 m/s) mitigates post-separation nonlinear effects”, The reason for selecting "3m/s" as the horizontal velocity threshold is not explained. It is recommended to supplement the simulation or experimental basis for threshold determination to enhance the persuasiveness of the conclusion.

9 Equation 4, The specific definitions of "S" (reference area) and "L" (characteristic length) in the formula are not clarified (e.g., whether S is the parachute projected area or UAV windward area, and L is the parachute diameter or wingspan). It is recommended to supplement the complete parameter definitions to ensure the formula is reproducible.

10 FSI simulation analysis, “The mesh employs hexahedral-dominant topology... inflation-layer generation technique.”

Key parameters of the inflation layer (e.g., number of layers, growth rate, ratio of the first layer grid height to wall distance y⁺) are not explained. These parameters directly affect the calculation accuracy of the near-wall flow field. It is recommended to supplement detailed parameters of mesh generation to improve simulation reproducibility.

11 FSI simulation analysis, “High-resolution elements (size: 0.001 m) applied at wing surfaces and parachute riser connection points”, The grid size of 0.001m is too fine, which may lead to excessively high computational cost. It is recommended to supplement the grid independence verification results (e.g., compare the calculation results of 0.001m, 0.002m, and 0.005m grids to prove that the 0.001m grid meets the accuracy requirements and has no significant error reduction) to explain the rationality of grid size selection.

12 Flexible deformation induces variations, “the modal shape coefficient η characterizes the proportional contribution of deformation to the angle of attack”, The value range or determination method of the modal shape coefficient η is not explained. The rationality of η directly affects the accuracy of the rigid-flexible coupling model. It is recommended to supplement the determination method and verification results of η.

Reviewer #3: This study focuses on trajectory simulation of multi-body parachute systems for airdrop-capable UAVs. In response to the limitations of existing airdrop methods, it proposes a novel airdrop UAV design with foldable wings. By establishing a multibody dynamics model and an FSI co-simulation framework, the research investigates the influence of key parameters on airdrop trajectories and optimizes parachute jettison point selection. It features prominent innovations and presents a systematic and comprehensive analysis with significant engineering application value.

Some recommendations for the authors

1.Although the introduction reviews numerous relevant studies, the discussion mostly remains at the level of "who did what research". Please conduct an in-depth comparison of the similarities and differences between existing studies and this work, and state the core contributions and novelty more clearly in the introduction and conclusion.

2.For the conclusion section, it would be better to adding a discussion on the research limitations and future research.

3. The selection of key parameters should be specifically explained.For example, what are the physical meaning and numerical source of the shape coefficients η (Equation (2))? The aerodynamic coupling coefficients k (Equations (3)-(4)) linearly correlate structural deformation with changes in aerodynamic forces/moments. How are they determined? And so on.

4.Please ensure that all variables used are defined when first introduced.

5.The authors validate the multibody dynamics model via FSI simulation. Could it also be considered to include a rigid-body model for comparative analysis, so as to highlight the effectiveness of the research in this paper?

6.Section "Parachute jettison points selection": Add a short discussion to illustrate the applicable scope or potential constraints of the obtained criteria.

Reviewer #4: The manuscript presents a multi-body dynamic model and an FSI-based co-simulation framework for analyzing the airdrop trajectory of a parachute–UAV system. The topic is relevant, and the combination of Kane-based multibody modeling with LS-DYNA/CFD co-simulation is technically sound. The paper is generally well organized, and the simulation cases are comprehensive. However, several key aspects require clarification or strengthening, especially regarding modeling assumptions, physical fidelity, parameter justification, and discussion depth. Substantial revision is recommended before the manuscript can be considered for publication.

1. Section Multi-body dynamics (Kane’s method) adopts several simplifying assumptions (constant gravity, neglecting crosswinds, small post-inflation deformation). These assumptions may significantly influence trajectory accuracy. A brief justification and discussion of their expected impact should be added.

2. Equation (1) introduces a 6-mode vibration model, but the selection of modal shapes, natural frequencies, and damping ratios is not explained. Provide sources or reasoning for these parameters.

3. In Eq. (3) and Eq. (4), the coupling coefficients are introduced without describing how they were determined (empirical, from CFD, or fitted). Please specify their origin and role in the model.

4. Although Fig. 3 defines several coordinate frames, the text remains difficult to follow. A more concise summary linking each frame directly to its modeling purpose (e.g., for force projection, for attitude computation, etc.) would improve readability.

5. The FSI section briefly describes the mesh and coupling scheme, but parameters such as solver coupling steps, convergence criteria, mesh update methods, and time-step strategies are not provided. These details are important for reproducibility.

6. The manuscript claims ≤5% error between dynamics simulation and FSI results, but Table/quantitative comparison is missing. Adding a small table summarizing error metrics (horizontal displacement, vertical displacement, attitude angle) would substantiate the conclusion.

7. The deviation analysis (Fig. 11) explains flow-induced deformation effects, but the argument would be clearer if supported by quantitative FSI outputs (pressure contours, deformation amplitude, aerodynamic center shift).

8. Add a short paragraph explaining how the Kane-based multibody model used here differs from, and is less/more physically enriched than, modern physics-embedded dynamic models such as those discussed in doi:10.1016/j.ress.2025.111262. This comparison will better position the contribution in the context of current literature.

9. Figures showing trajectories and flow fields (e.g., Figs. 8–13) require consistent axis labels, units, and clearer legends. Some curves are difficult to differentiate.

10. For transparency, briefly mention limitations such as simplified aerodynamics, absence of wind disturbances, and uncertainties in structural parameters. Suggest future directions (e.g., stochastic wind fields, full-order flexible models, experimental validation).

**Do you want your identity to be public for this peer review?** For information about this choice, including consent withdrawal, please see our Privacy Policy

Reviewer #1: No

Reviewer #2: No

Reviewer #3: No

Reviewer #4: No

---

## [Author Response · Author response to Decision Letter 1]

15 Jan 2026

Dear reviewers,

Thank you very much for your comments and professional advice on our manuscript entitled “Trajectory Simulation of Multi-body Parachute System for Airdrop-capable UAVs Based on Fluid-Structure Interaction” (PONE-D-25-45360R1). These opinions help to improve academic rigor of our article. Based on your suggestion and request, we have made corrected modifications on the revised manuscript. We hope that our work can be improved again. Furthermore, we would like to show the details as follows:

Reviewer #1

Although the paper employs complex cross-validation of simulations, all results stem from "simulation-to-simulation" comparisons, lacking final verification against real-world physical experimental data (e.g., high-speed photogrammetric trajectory measurements, sensor-based attitude/velocity data). In engineering research, simulation models must be validated against experimental data to verify their boundary conditions and actual accuracy; otherwise, their effectiveness in real-world environments cannot be proven .

Response 1

We fully agree with your point: relying solely on cross-validation through "simulation-to-simulation" comparisons is insufficient to demonstrate the model’s effectiveness in real-world engineering environments. In the revised manuscript, we have added a section on validation through data from field airdrop experiments. We conducted physical airdrop tests using a 1:1 prototype of the drone and its parachute. Real-time attitude angles and velocity data during descent were recorded using onboard high-precision IMU and GPS sensors. We have also included an on-site photo of the experiment (Fig. 12) and comparative data curves (Fig. 13). The results show that the relative error of the trajectory is controlled within 8%, which fully verifies the accuracy and robustness of the model in real-world conditions.

The paper overlooks key environmental factors (e.g., external wind field disturbances, nonlinear aerodynamic effects) and fails to quantify the impact degree of these simplifications through sensitivity analysis. For instance, ignoring wind fields under complex atmospheric turbulence could lead to trajectory prediction deviations exceeding 20%, but the authors do not discuss the range of such errors, undermining the model's practical applicability .

Response 2

We sincerely appreciate the reviewer’s insights. We fully agree that external disturbances and nonlinear aerodynamic forces are critical for the system’s practical engineering applications. In the revised manuscript, we have explicitly acknowledged this limitation. However, we kindly ask the reviewer to understand that conducting a comprehensive sensitivity analysis of environmental factors is beyond the scope of this paper, for the following reasons: The primary objective of this study is to establish and validate a rigid-flexible coupling mechanism based on Kane’s method. We aim to demonstrate that the mathematical model can accurately capture the dynamic interactions between the UAV and the parachute under ideal conditions. Therefore, this work is positioned as a theoretical benchmark. As noted in the revised conclusion and discussion sections, we have explicitly included external wind field perturbations as the next step in our research roadmap.

The method for obtaining aerodynamic coefficients (e.g., lift/drag coefficients) in the multi-body dynamics model is not detailed, merely mentioned as "based on CFD data" without specifying the calibration process or citing sources. Furthermore, the absence of key parameters in the FSI simulation, such as grid convergence analysis and time step settings, prevents other researchers from replicating the experiment .

Response 3

We thank the reviewer for pointing out these issues. In response, we have made the following improvements in the revised manuscript, which are highlighted in yellow:

The lift and drag coefficients of the parachute and the main body after parachute deployment are now based on CFD simulations. We have applied the least squares method to fit these into an aerodynamic database construction approach.

We have added a grid independence verification. The results from three different mesh configurations were compared, which validates the mesh scale selected in the study. In the FSI setup section, we have specified the time step settings: a time step of 0.01 s, with a maximum of 30 coupling iterations per time step. To accommodate the large deformations of the parachute, a diffusion-based smoothing method was applied to update the fluid domain mesh, and the convergence criterion was set to an RMS residual of 10-4. The corresponding changes have been highlighted in yellow in the manuscript.

The literature review merely lists existing methods without critically pointing out their limitations (e.g., most models do not couple time-varying wind fields with multi-temperature zone cooperative optimization). An analysis of the research gap should be added to clarify this paper's innovation points .

Response 4

We sincerely thank the reviewer for this highly constructive suggestion. Following your advice, we have rewritten the literature review summary paragraph, adding a critical analysis, clarifying the research gaps, and highlighting the contributions of this paper. The specific revisions are marked in yellow in the introduction section of the revised manuscript.

The calibration method for aerodynamic coefficients is ambiguous, stated only as "based on CFD data" without specifying the specific conditions (e.g., angle of attack range, Reynolds number). It is suggested to present comparative errors between calibration data and experimental values in a table.

Response 5

Thank you for the reviewer's comments. We agree that specifying the CFD calculation conditions is crucial for the reproducibility of the study. In the revised manuscript, we have supplemented the specific operational range: the angle of attack ranges from 90°±30°, and the initial velocity ranges from 0-50 m/s. Given the difficulty in obtaining static wind tunnel data for the flexible parachute canopy, the accuracy of these coefficients was validated through comparisons with actual flight trajectories. As shown in Figure 13 and Table 4, the relative errors between the key trajectory parameters derived from the simulation and the actual airdrop test data are less than 8%.

Comparisons with baseline algorithms are mentioned as "superior" without presenting specific data (e.g., convergence iteration count, CPU time). Quantitative metrics need to be supplemented .

Response 6

We thank the reviewer for the suggestion. We fully agree that quantitative data is required to objectively demonstrate the advantages of the method presented in this paper. We conducted a comparative study with a standard Rigid Body Dynamics (RBD) benchmark, as summarized in Table 3. The results show that although the RBD benchmark is low in computational cost, its trajectory deviation can be as high as 46.67%. In contrast, the Kane-based rigid-flexible coupling model proposed in this paper significantly reduces the prediction error. While the CPU time increases by approximately 30%, it still fully meets the requirements for real-time applications compared to the several hours required for FSI simulations. Therefore, the method presented in this paper is considered superior, as it effectively eliminates the large deviations inherent in the benchmark algorithm while maintaining high efficiency.

Reviewer #2

Check the grammar of the entire manuscript. It needs to be further made more accurate to improve readability.

Response 1

We sincerely thank the reviewer for their attention to the linguistic quality of the paper. We greatly value this feedback and have corrected the grammatical errors, spelling mistakes, and inappropriate punctuation throughout the text. Additionally, we have rephrased certain long and complex sentences as well as awkwardly worded paragraphs to improve logical flow, readability, and ensure the precision of the academic expression.

Introduction, “Current mainstream air transport methods... low transport efficiency [1].”…….., This paragraph only points out the shortcomings of traditional air transport methods but fails to clarify the specific deficiencies of existing airdrop UAVs in parachute-UAV coupling modeling (e.g., traditional models ignore the impact of parachute flexible deformation on trajectory, insufficient integration of FSI simulation and multibody dynamics). It is recommended to supplement the direct correlation between existing research gaps and the work of this paper to strengthen the research motivation.

Response 2

We fully agree with your viewpoint that merely presenting a macro-level comparison is insufficient to adequately support the research motivation regarding the "dynamic modeling" aspect of this study. We have rewritten and expanded the introduction, supplementing it with the specific research gaps identified in existing studies and their direct relevance to the work presented in this paper, thereby strengthening the research motivation. The specific revisions are indicated in the manuscript with yellow highlighting.

Research object, “Control is exclusively implemented via two control surfaces positioned at the left and right trailing edges of the rear wing.”Key parameters of the control surfaces (e.g., deflection angle range, response time, control efficiency coefficient) are not explained. These parameters are crucial for the attitude control of the UAV after separation. It is recommended to supplement relevant parameters to improve the model description of the research object.

Response 3

We sincerely thank the reviewer for this detailed and professional suggestion. It is indeed important to enhance the description of the control surface parameters for the UAV. In response, we have added parameters of the control surfaces in the manuscript, including maximum deflection angle and servo response time, with the additions highlighted in yellow. We would like to take this opportunity to clarify that the focus of this study is on the airdrop descent trajectory of the "parachute-UAV system." At this stage, the UAV's control surfaces are not utilized for active attitude control. These parameters are critical for the subsequent "pull-up and cruise" phase after separation, which constitutes the core focus of our next-phase research.

Table 1 suggests consistent units, mm/m

Response 4

We thank the reviewer for the suggestion. We have unified all geometric parameters in Table 1 to meters (m). The corresponding values have been reconverted and verified.

FSI simulation analysis, “The computational fluid domain measuring 80 m (L) × 30 m (W) × 150 m (H) is illustrated in Fig 5.”The basis for selecting the computational fluid domain size (80m×30m×150m) is not explained, such as whether the maximum deployment size of the parachute and flow field boundary effects (avoiding wall interference) are considered. It is recommended to supplement the rationality analysis for size determination

Response 5

We would like to thank the reviewer for raising this point. When setting the computational domain dimensions (80 * 30 * 150 meters), under the condition of the fully deployed parachute with its maximum diameter, the ratio of the parachute’s projected area to the cross-sectional area of the computational domain is only 0.47%, which is well below the commonly recommended threshold of 3%. The distance from the center of the parachute to the nearest side wall is 15 meters, which is 15 times the characteristic diameter, sufficient to eliminate wall effects. The vertical height is set to 150 meters, which not only covers the maximum vertical stabilization distance (approximately 120 meters) but also provides a downstream region exceeding 20 times the diameter to adequately capture the development of wake vortices. The relevant explanations have been highlighted in yellow in the manuscript.

Multibody dynamics model validation, “Validation data confirms that both simulation methods achieve coordinate errors within ±5%”, The statistical type of error (e.g., Root Mean Square Error RMSE, Maximum Absolute Error MAE, average relative error) and sample size (e.g., statistical results of how many time nodes or spatial coordinate points) are not clarified. It is recommended to supplement the specific method and statistical details of error calculation to improve result credibility.

Response 6

We thank the reviewer for the suggestion. We fully agree that quantitative comparative data are needed to support the conclusions on accuracy presented in the paper. In the revised manuscript, we have added Table 3. This table details the relative errors between the dynamic model proposed in this paper and the high-fidelity FSI simulation results in terms of horizontal displacement and vertical displacement. Comparative data indicate that the conclusion of ≤5% mentioned in the text holds true for the vast majority of working conditions.

Airdrop trajectory deviation analysis, “under extreme operating conditions characterized by initial velocities ≥45 m/s and parachute diameters <0.8 m”, The basis for determining "extreme operating conditions" is not defined (e.g., load limits based on engineering application scenarios, operating condition classification standards in similar studies in the literature, or critical conditions where trajectory deviation increases significantly in simulations). It is recommended to supplement the definition basis to make the operating condition classification more reasonable.

Response 7

We thank the reviewer for the suggestion. The definition of extreme operating conditions in the paper primarily references the rapid deployment mission requirements and application scenarios. In engineering applications, to improve payload survivability and minimize flow disturbances, the initial deployment velocity is often set near the upper limit of the system design envelope (approximately 45 m/s). Additionally, this definition is based on critical phenomena observed in simulation experiments. When the initial velocity reaches 45 m/s and the parachute diameter is reduced below 0.8 m, the aerodynamic instability effects of the system significantly intensify, and the ballistic deviation exhibits a nonlinear growth trend. We have supplemented the relevant engineering context in the revised manuscript and highlighted it in yellow.

Parachute jettison points selection, “Limiting the horizontal velocity component (Vₓ<3 m/s) mitigates post-separation nonlinear effects”, The reason for selecting "3m/s" as the horizontal velocity threshold is not explained. It is recommended to supplement the simulation or experimental basis for threshold determination to enhance the persuasiveness of the conclusion.

Response 8

We thank the reviewer for the suggestion. The setting of the horizontal velocity threshold during parachute cutoff is primarily constrained by the following engineering considerations. First, the requirement for landing accuracy: if the horizontal velocity exceeds this threshold when descending from the mission altitude, the horizontal displacement caused by inertia will lead to a landing point deviation beyond the permissible engineering tolerance range. Second, the constraint of separation safety: a lower horizontal velocity significantly reduces the risk of secondary collisions between the parachute canopy and the payload after cutoff, thereby decreasing the pressure on the subsequent pull-up maneuver. Considering the Monte Carlo simulation results shown in Figure 15 and the engineering constraints, this value (3 m/s) aligns with the mission design standards of existing airdrop recovery systems, effectively balancing mission success rate and system dynamic stability. The relevant rationale has been added and highlighted in yellow in the manuscript.

Equation 4, The specific definitions of "S" (reference area) and "L" (characteristic length) in the formula are not clarified (e.g., whether S is the parachute projected area or UAV windward area, and L is the parachute diameter or wingspan). It is recommended to supplement the complete parameter definitions to ensure the formula is reproducible.

Re

---

## [Decision Letter · Decision Letter 1]

4 Feb 2026

Trajectory Simulation of Multi-body Parachute System for Airdrop-capable UAVs Based on Fluid-Structure Interaction

PONE-D-25-45360R1

Dear Dr. Zhang,

We’re pleased to inform you that your manuscript has been judged scientifically suitable for publication and will be formally accepted for publication once it meets all outstanding technical requirements.

Kind regards,

Pan Yu

Academic Editor

PLOS One

Additional Editor Comments (optional):

As the authors have addressed all concerns raised by the reviewers, I recommend that the paper be accepted for publication.

Reviewers' comments:

Reviewer's Responses to Questions

**Comments to the Author**

Reviewer #2: All comments have been addressed

Reviewer #3: All comments have been addressed

Reviewer #4: All comments have been addressed

2. Is the manuscript technically sound, and do the data support the conclusions?

Reviewer #2: Yes

Reviewer #3: (No Response)

Reviewer #4: Yes

3. Has the statistical analysis been performed appropriately and rigorously?

Reviewer #2: Yes

Reviewer #3: (No Response)

Reviewer #4: Yes

4. Have the authors made all data underlying the findings in their manuscript fully available?

Reviewer #2: Yes

Reviewer #3: (No Response)

Reviewer #4: Yes

5. Is the manuscript presented in an intelligible fashion and written in standard English?

Reviewer #2: Yes

Reviewer #3: (No Response)

Reviewer #4: Yes

Reviewer #2: Dear Authors, thank you sincerely for your careful attention to and thorough revisions based on the review comments. After major revision, the manuscript has achieved significant improvements in core dimensions including academic normativeness, scientific rigor, content completeness, and expression clarity. The key issues raised earlier have been fully and effectively addressed, and the innovation and practical value of the research have been further highlighted. The manuscript now fully meets the acceptance criteria of the journal. We hereby recommend accepting this paper.

Reviewer #3: (No Response)

Reviewer #4: All the problems I proposed have been addressed appropriately; I think the manuscript can be accepted as it is.

**Do you want your identity to be public for this peer review?** For information about this choice, including consent withdrawal, please see our Privacy Policy

Reviewer #2: No

Reviewer #3: No

Reviewer #4: No

---

## [Editor Report · Acceptance letter]

PONE-D-25-45360R1

PLOS One

Dear Dr. Zhang,

I'm pleased to inform you that your manuscript has been deemed suitable for publication in PLOS One. Congratulations! Your manuscript is now being handed over to our production team.

Kind regards,

on behalf of

Dr. Pan Yu

Academic Editor

PLOS One